# A 4,000 year debris-flow record based on amphibious investigations of fan delta activity in Plansee (Austria, Eastern Alps)

Carolin Kiefer*[1], Patrick Oswald*[2], Jasper Moernaut[2], Stefano Claudio Fabbri[3], Christoph Mayr[4], Michael Strasser[2], Michael Krautblatter[1],

5    *These authors contributed equally to this work.

[1]Chair of Landslide Research, Technical University of Munich, Arcisstraße 21, 80333 Munich, Germany

[2]Department of Geology, University of Innsbruck, Innrain 52 f, 6020 Innsbruck, Austria

[3]Institute of Geological Sciences and Oeschger Centre for Climate Change Research, University of Bern, Baltzerstrasse 1+3, 3012 Bern, Switzerland

10   [4]Institute of Geography, Friedrich-Alexander University Erlangen-Nuremberg, Glückstraße 5, 91054 Erlangen, Germany

*Correspondence to*: Carolin Kiefer (Carolin.Kiefer@tum.de)

**Abstract**

The frequency of debris flows is hypothesized to have increased in recent decades with enhanced rainstorm activity. Geological evidence to test the relationship between climate and debris flow activity for prehistoric times is scarce due to incomplete sediment records, complex stratigraphy, and insufficient age control especially in Alpine environments. In lacustrine archives, the link between onshore debris-flow processes and the sedimentary record in lakes is poorly investigated. We present an amphibious characterization of alluvial fan deltas and a continuous 4,000 year debris-flow record from Plansee (Tyrol, Austria) combining Light Detection And Ranging (LiDAR) data, swath bathymetry, and sediment core analyses. The geomorphic investigation of two fan deltas in different developmental stages revealed an evolutionary pattern of backfilling and new channel formation onshore together with active subaqueous progradation on a juvenile fan delta, major onshore sediment deposition and only few but larger subaqueous deposits on a mature fan delta. Geomorphic evidence for stacked and braided debris-flow lobes, subaquatic landslide deposits, and different types of turbidites in sediment cores facilitated a process-based event identification i.e. distinguishing between debris-flow or earthquake-induced turbidites throughout the 4,000 year sedimentary record. We directly correlate subaqueous lobe-shaped deposits with high backscatter signals to terrestrial debris-flow activity of the last century. Moreover, turbidite thickness distribution along a transect of four cores allows us to pinpoint numerous events as being related to debris-flow activity on a juvenile fan delta. In the sediment core, debris flow-induced turbidites feature a more gradual fining-upward grain-size trend and higher TOC and $\delta^{13}$C values compared to earthquake-induced turbidites. The 4,000 year event record contains 138 debris flow-induced turbidites separated into four phases of similar debris-flow activity (df phases). Df phase 1 (~2120 to ~2040 before the common era; BCE) reflects the second highest observed event frequencies and is interpreted as postseismic landscape response. After a long period of long recurrence intervals without any outstanding increases in debris-flow activity during df phase 2 (~2040 BCE to ~1520 common era; CE) there are slightly increased event frequencies in df phase 3 (~1520 to ~1920 CE). Df phase 4 (~1920 to 2018 CE) exhibits a drastic increase in debris-flow activity followed by the overall highest debris-flow frequency of the whole record, which is about 7 times higher than during df phase 3. We show that the frequency increase in the debris flow-induced turbidite record matches a previously postulated increase in debris-flow events derived from aerial photography at Plansee in the last century. The triggering of debris flows is more controlled by short intense precipitation than any other mass movement process and we demonstrate that lacustrine debris-flow records provide a unique inventory of hazard-relevant rainstorm frequencies over decades, centuries, and millennia. The presented increase in debris-flow frequency since the 20th century coincides with a 2-fold enhanced rainstorm activity in the Northern European Alps and therefore provides a novel technique for systematic understanding of non-stationary debris-flow frequencies in a changing climate.

**1 Introduction**

Debris flows are among the most important hazards in alpine geosystems and are responsible for ca. 10,000 casualties per decade worldwide (Dowling and Santi, 2014). They represent some of the most hazardous mass movements due to their highly destructive combination of hydrodynamic pressure, hydrostatic pressure, and collisional forces (Thouret et al., 2020). Most debris flows commence as landslides triggered by increased pore water pressures, and most terminate as slowly consolidating sediment deposits (Iverson, 1997). Climate change influences debris-flow frequencies and magnitudes through massive debris release in periglacial high mountain settings due to glacier retreat and permafrost degradation (Damm and Felderer, 2013; Chiarle et al., 2007) and more generally due to enhanced rainstorm activity in the last century (Dietrich and Krautblatter, 2017). Climatic warming-induced glacier retreat and the degradation of permafrost during the 19th and 20th century favored debris-flow activity (Zimmermann et al., 1997). Enhanced sediment production in permafrost-affected altitudes and periglacial settings produces massively elevated debris-flow activity, and the most affected altitude range is projected to extend upwards in the coming decades (Pavlova et al., 2014; Jomelli et al., 2009). More generally, alterations in the intensity and duration of

short-term precipitation control debris-flow activity in all altitudes. Regional studies in the French Alps show that >70 % of all debris flows can directly be attributed to intense precipitation (Jomelli et al., 2019). In many mountain environments worldwide, the number of extreme rainfall events capable of triggering debris flows in the summer months has increased in the 20[th] century (e.g. Rebetez et al., 1997). Blöschl et al. (2020) showed that the past 3 decades were among the most flood-rich periods in Europe in the past 500 years. Many of the largest debris flows in the Alps in the past 20 years were triggered by intense rainfall in summer or fall when the snowline was elevated (Rickenmann and Zimmermann, 1993). Schlögel et al. (2020) investigated in situ and satellite-based climate data in South Tyrol and observed an increase in the average annual duration of rainfall events (+1.1 hours per year) and debris-flow occurrence (+1.2 events per year from 1998 to 2018). Other studies expect an increasing magnitude of debris flows due to an increased availability of loose sediment, longer return periods and presumably fewer, but more intense rainfalls in summer (Stoffel et al., 2014; Stoffel, 2010).

The frequency of debris flows over longer time scales is difficult to estimate as terrestrial inventories rarely provide stratigraphically distinct and continuous evidence of subsequent debris flows. There is a lack of continuous long-term datasets to evaluate if Central Europe is in a period of high debris-flow activity compared to the preceding millennia and if long term variations in debris-flow activity follow climatic trends (Stoffel et al., 2005; Irmler et al., 2006). Obtaining such regional perspective is especially challenging due to the local imprint of debris flows and a potentially local trigger process, such as convective storms. The investigation of debris-flow frequencies and magnitudes in lacustrine environments may provide reliable data on prehistorical changes because the typical continuous sedimentation regime can lead to a complete high-resolution archive in which evidence for individual debris flows is preserved (Irmler et al., 2006).

Despite being widely used for reconstructing past river flood activity (e.g. Gilli et al., 2013; Schillereff et al., 2014; Wilhelm et al., 2019), lacustrine sediments remain an underexplored archive for debris-flow studies. This results from three challenges related to these lacustrine inventories: (i) debris flows are in many studies not distinguished from other sources of coarser grained sediments such as floods and landslides outside and inside the lake, (ii) debris-flow volumes could not be quantitatively assessed as only lake bottom sequences in sediment cores were analysed, and (iii) there is a lack of long-term instrumental debris-flow data that covers the period with less human interference. To address these challenges, we here combine on- and offshore investigations to identify and reconstruct debris-flow dynamics at Plansee, a mid-elevation lake in the Alps, that acts as a natural, continuous sedimentary archive (Oswald et al., 2021). For a conclusive identification of debris-flow turbidites in the sediment core, we map geomorphic landforms in both the subaquatic and terrestrial realms, which document the interplay between the terrestrial source area, terrestrial and subaqueous sediment transport, potential temporary storage on the fan delta and the final sink in the depocentre.

An increase of debris-flow activity in the Plansee catchment was previously derived from aerial photography by Dietrich and Krautblatter (2017). Mean debris-flow rates on eight investigated fans have increased by a factor of more than three since the 1980s in comparison to a reference period from the 1940s to 1970s. This may link to the doubled frequency of heavy rainfall events (≥35 mm d$^{-1}$) from 1920 to 2010 CE in the Plansee area (Dietrich and Krautblatter, 2017; Hydrographischer Dienst Tirol, 2020).

In this study, two types of alluvial fan deltas are investigated at Plansee, representing the end members of premature and fully developed geomorphological evolution, hereafter referred to as "juvenile" and "mature" fan deltas, respectively. A juvenile fan delta is a high-sloping (>20°) semi-conical shaped deposit adjacent to a trough-like channel cross-section. In this early stage of fan development, the deposition of material reduces the capacity of the channel, so shifts in the channel course resulting in interfingering deposits are likely to occur over time, therefore gradually building up the semi-conical shape (e.g. Sass and Krautblatter, 2007). A mature fan delta displays a late stage of fan development, where the morphology has flattened out following a long period of sediment delivery and the terrestrial profile extends into the lake. It is connected to a large catchment and shows fluvial influence. The large delta depicts a lobate-shaped fan and lower mean slope angles compared to the juvenile fan delta.

Instantaneous deposits are a major contributor to lacustrine sedimentation in Plansee (Oswald et al., 2021). The largest share of sediment transport into the lake occurs at the alluvial fans in the form of debris flows, which incorporate more water and turn into a turbulent high-density current when entering the lake (Lowe, 1982). The resulting sediment deposits at the lake bottom are referred to as "df turbidites" hereafter.

We investigate subaerial and subaquatic sediment dynamics of the juvenile and mature debris-flow system and tackle the following research questions: (i) How can df turbidites be distinguished from other sediment sources in inner-alpine lakes? (ii) What is the ratio of terrestrial and subaqueous deposition of recent debris flows on juvenile and mature fan deltas? (iii) How are the geomorphic expressions of debris flows related in terrestrial digital elevation model (DEM) and bathymetry data? (iv) Can we systematize the subaqueous deposition pattern of debris flows? (v) Can we decipher the frequency in the last millennia to reveal the recent peak activity of debris flows?

## 2 Study site

The inner-alpine lake Plansee (surface area: 2.78 km$^2$; maximum depth: 78 m) is located in the Northern Calcareous Alps in Austria, North Tyrol (47°28'10" N, 10°48'00" E; 976 meter above sea level, m a.s.l.) and was formed in a glaciated alpine trough valley. The lake is surrounded by numerous alluvial fans, talus slopes and fan deltas, which subdivide it into two main basins (Fig. 1). In the SW, Plansee and Heiterwangersee, originally separated by a large fan delta, were connected by a 300 m long canal in 1908 (Hibler, 1921). The lake has two permanent river inflows in the eastern basin, several inflowing ephemeral streams on the alluvial fans and one outflow in the NW. The herein investigated main (westernmost) basin of Plansee is isolated by circa 20 m high morphological barriers from the circa 4 km distant permanent river inflows in the east, which protects this basin from the influence of river flood-induced turbidity currents. Therefore, the main basin forms a rather unique setting to study debris flows and their related subaqueous deposits, as it is expected that detrital sediment is almost exclusively supplied by episodically occurring debris flows. Since 1902, the lake is used as a reservoir for hydropower generation causing artificial lake level fluctuations of up to 5 m in the winter (Hydrographischer Dienst Tirol, 2020). This study is focused on episodic debris flows and their related lacustrine deposits. Human interference on the volumes of these debris flows only affects a few percent of the contributing catchments and is limited to the lower depositional domain of the debris-flow channels. While the human interference may influence the continuous background sedimentation in the lake, the episodic debris flows eroding materials from 10,000 m² large, steep catchments way above the lake will not be relevantly influenced.

The mountains surrounding Plansee consist of intensely jointed (cm to dm scale) Upper Triassic lagoonal dolomites (Hauptdolomit), mechanical erosion of which provides a vast amount of loose sediment to the upper catchment areas, which can be remobilized during extreme precipitation events, e.g. in the form of debris flows along incised ditches and canyons. The slopes in the direct vicinity of the lake are dominated by numerous fan deltas of different developmental stages and subordinated talus slopes (Fig. 1). The fan deltas overlie a local glacial till and reach up to 25 m thickness near the lake shore (Dietrich and Krautblatter, 2017), which highlights the vast amount of remobilized sediment derived from only small and local catchments with areas ranging from ~0.05 to 1.5 km². The steep forested slopes are prone to episodic gravitational mass-transport processes propagating into the lake.

Previous work onshore Plansee investigated a debris-flow fan with electrical resistivity tomography combined with orthophoto analysis of the last century (Fig. 1d) and linked increased debris-flow volumes since the 1980s to enhanced rainstorm activity in the study area (Dietrich and Krautblatter, 2017). The question arises of whether such an increase in debris-flow activity can be validated in a continuous millennial-scale lacustrine record. Previous limnogeological work on Plansee investigated subaquatic mass-wasting events recorded in the stratigraphy based on subbottom profiles and a 7 m long sediment core in the main basin, and inferred five severe Holocene earthquakes (local magnitude $M_L \geq 5.3$; Oswald et al., 2021). Furthermore, the sedimentary sequence in the main basin of Plansee can also archive extreme sediment transport events in high resolution and

high continuity, which sets the stage for the herein presented study on the identification of debris flow-related turbidites and their frequency over time.

140 The annual mean precipitation is 1,700 mm in the study area with a pronounced summer rainstorm precipitation maximum obtained from two nearby meteorological stations recording since 1900 CE (Höfen, Berwang; Hydrographischer Dienst Tirol, 2020). The relative frequency of heavy rainfall events ($\geq$35 mm d$^{-1}$) at the nearest meteorological station "Berwang" has increased on average by 10 % per decade from 1920 to 2010 CE (Hydrographischer Dienst Tirol, 2020) raising the hypothesis of also increased debris-flow activity since then. Two extraordinary large cyclonic rainstorms with overall damage of hundreds

145 of million US$ in May 1999 and August 2005 CE (Barredo, 2007) also hit the study area with a peak daily sum precipitation of 180 mm and 130 mm, respectively, measured at the weather station in Berwang ~7 km SE of Plansee (Hydrographischer Dienst Tirol, 2020).

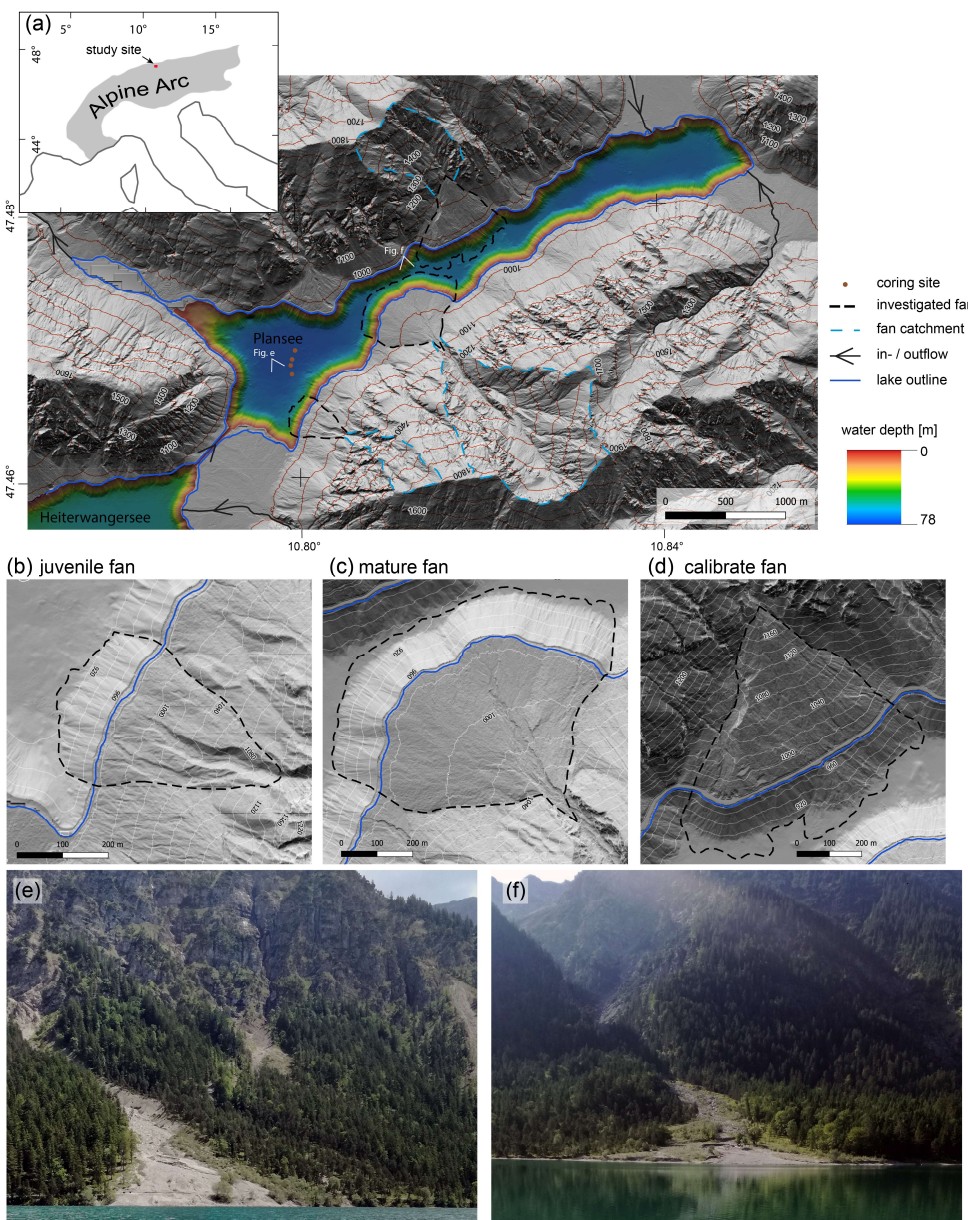

**Fig. 1: Overview map of Plansee and the investigated alluvial fan deltas. a) Bathymetry of Plansee highlighting the investigated fan deltas (dashed black lines) with their catchments (blue dashed lines) and coring sites (red dots) used for characterization of subaquatic debris-flow turbidites. Upper left inlet indicates location of Plansee within the Alpine arc. Hillshades of combined DEM and bathymetry of b) the small and steep juvenile fan, c) the large and low angle mature fan, and d) average-sized fan delta with documented debris-flow activity in 1947–2014 CE (Dietrich and Krautblatter, 2017) used for characterization of geomorphic features. The juvenile (b, e) and mature (c, f) fans were repeatedly investigated by terrestrial laser scan measurements in 2019 (see Fig. 4, 5). Onshore DEM derived from Land Tirol (data.tirol.gv.at).**

## 3 Data and methods

### 3.1 Conceptual approach

The coupled study of debris-flow systems on land and underwater delivers new insights into geomorphic expressions from catchment to depocentre. An amphibious geomorphological investigation allows us to assess the influence of catchment conditions, delta dynamics, and deposition patterns and to define additional identification criteria for different processes in the event stratigraphy. The determination of geomorphic landforms, i.e. debris-flow deposits quantitatively onshore on a seasonal time scale, and qualitatively on the subaquatic slope, gives a general understanding of the system connectivity. Differences in the subaquatic distribution of deposits due to catchment and delta characteristics can be identified and the selection of a transect and coring site can be justified (see Sletten et al., 2003; Irmler et al., 2006).

Based on the geomorphological investigations two representative types of fan deltas on the southern shore of the lake—a juvenile, steep debris cone and a mature, low angle fan delta (Fig. 1b, c)—were chosen for further investigations by repeated terrestrial laser scanning (TLS) measurements and a multibeam bathymetric survey in 2019. TLS offers a precise method for quantifying short-term volume changes on the terrestrial part of alluvial fan deltas, while bathymetric surveys reveal the subaquatic continuation of fan deltas and subaquatic debris-flow deposition patterns, and provide hints about the present grain size at the lake floor. Furthermore, a transect of four ~1.5 m sediment cores from the juvenile fan delta towards the 78 m deep depocentre of the main basin provides the means for deriving the relative thickness distribution of turbidites from the subaqueous delta slope to the basin. A combination of short-lived radionuclides and radiocarbon ages of the sedimentary succession is used to establish an age–depth model and to date the individual event deposits. In this study, (i) onshore debris-flow volumes are calculated, (ii) two types of lacustrine event deposits are differentiated, iii) the spatial extent of subaquatic high-density currents is determined, and (iv) long-term sedimentation patterns are analysed.

### 3.2 Terrestrial LiDAR (Light Detection And Ranging) data acquisition and processing

Topographic surveying of debris-flow volumes was conducted on two alluvial fan deltas bordering the lake. Two digital terrain models derived from consecutive TLS were compared for each fan delta. Our premise derived from field observation is that the LiDAR covered area of the fan includes the major proportion of redistributed sediments since the higher inaccessible steep catchment parts cannot accumulate significant amounts of debris. Computing DEM difference rasters is a straightforward and commonly applied method to detect topographic surface changes (Bremer and Sass, 2012; Abellán et al., 2009). The fan deltas were scanned from five scan positions on May 10th and August 22nd 2019 using a Riegl VZ-400 laser scanner (long range mode, near-infrared wavelength, measurement range 1.5–600 m, accuracy 5 mm, precision 3 mm, measurement rate 42,000 pts s$^{-1}$, beam divergence 0.3 mrad; RIEGL Laser Measurement Systems GmbH, 2017). Data processing of the point clouds was executed with RiSCAN Pro (v. 2.9). After a coarse error removal, all point clouds of a survey date were coarsely registered with four corresponding points in two consecutive scans and were then fine registered by a multi station adjustment, which uses planar patches of the point clouds, resulting in a 3.1 cm mean deviation of the 3D distances. Vegetation was automatically eliminated by filtering the point cloud with a multidimensional terrain filter, followed by manual removal of remaining shrubs. A 2.5 D digital terrain model was derived by triangulation. Distances between the May and August models were measured perpendicular to the XY plane using the 'surface comparison' tool. The resulting point cloud was rasterized with the software Cloud Compare (v. 2.11.0; Grid step 0.12; average cell height and scalar field values). The differential volumes (erosion and deposition) over the 3-month period were calculated by creating volumetric meshes with RiSCAN Pro. The Sediment Delivery Ratio (SDR) is an indicator for the proportional sedimentation on land and in the lake. It was calculated following Eq. (1) (Lu et al., 2006), where the sediment flux into the lake is the difference between erosion and deposition volume.

$$(1) \; SDR = \frac{Sediment \; Flux \; into \; the \; Lake \; [m^3]}{Erosion \; Volume \; [m^3]}$$

### 3.3 Limnogeological data

#### 3.3.1 Swath bathymetry

High-resolution bathymetry data was acquired in October 2019 by a Kongsberg EM2040 multibeam echo sounder (University of Bern) operating at 300 kHz in a 1 by 1 degree beam-width configuration. For positioning, a Leica GX1230+ GNSS receiver was used in combination with real-time kinematic corrections (RTK) provided by EPOSA (real-time positioning service Austria, EPOSA, 2021). Sound correction was based on continuously monitored surface sound-velocity and nine vertical velocity profiles recorded at least twice per day using a Valeport MiniSVP probe. Speed of sound in the water column ranged from ~1,452 m s$^{-1}$ at the surface to 1,424 m s$^{-1}$ in the deepest basin, and 1,446 m s$^{-1}$ to 1,428 m s$^{-1}$ in Heiterwangersee. The recorded raw data have been processed in Caris HIPS/SIPS 9.1 software. During processing, all auxiliary sensor data (motion sensor, heading sensor, GNSS sensor) are merged, reviewed and manually corrected if necessary. Daily lake level changes (4–6 cm in 24 hours with respect to a reference level of 976.0 m a.s.l.) were corrected using data from a local gauging station (E-Werke Plansee). The resulting point cloud (~200 million points) was reviewed and different algorithms for rasterizations were tested, resulting in a bathymetric map with 1 m horizontal and a few decimetres vertical resolution. Besides depth information, the amplitude of the backscattered acoustic signal was calculated based on a median temperature of 4.7° C, a median sound velocity of 1,425 m s$^{-1}$ and an assumed salinity of 0 ppt from a vertical sound velocity profile taken in the deepest part of the lake. Backscatter data provide rough estimates on the sediment grain size by coarser grain sizes yielding higher amplitude values (Beyer et al., 2007; Hilbe et al., 2011). Topographic openness maps were calculated with SAGA GIS and combined with analytical hillshades for enhanced visualisation of the geomorphic features (Fig. 2) or combined with a color-ramped shading representing the water depth (Fig. 1) using QGIS Desktop 3.10.5 (see also Daxer et al. (2020) and references therein for a detailed description). Interpretation of subaquatic geomorphic landforms was carried out following Strasser et al. (2020) and references therein.

#### 3.3.2 Sediment core analyses

Four ~1.5 m long sediment cores with 63 mm diameter were retrieved in 2018 and 2019 CE using a gravity coring system equipped with a manual percussion system (Table S1). Preceding evaluation of recent fan delta activity in combined LiDAR and bathymetric data defined a suitable fan delta from which the core transect was taken towards the distal depocentre at intervals of 50-75 m (Fig. 1). The selected coring sites are located far from the slope break to minimize potential erosion (Fig. S1). In the lab, sediment cores were split lengthwise, macroscopically described, imaged using a Smartcube® Camera Image Scanner and scanned for petrophysical properties using a GEOTEK Multi Sensor Core logger. A core-to-core stratigraphic correlation was conducted based on sedimentary facies, density, lightness (L*) and distinct marker layers. Event deposits of ≥1 mm thickness were identified directly on the core surface in combination with color- and contrast-enhanced core images (Automatic Histogram Equalization).

We macroscopically characterized, mapped, and correlated the event deposits in all four sediment cores following the sedimentological criteria outlined in Sect. 3.3.3 and based on Irmler et al. (2006), and measured the thickness of each event deposit. Event deposit thicknesses of the uppermost sediments were corrected based on water content for comparable measurements with deeper deposits with lower water content.

Laser-diffraction grain-size analyses were performed on event deposits using a Malvern Mastersizer 3,000 in combination with a Hydro Sight module for visualization and quality control of the dispersion. Samples were measured without any chemical pre-treatment, as the amount of organic material is negligible in the clastic-dominated sediments. Samples were taken using a toothpick with a resolution of up to 2 mm dependent on deposit thickness. The measurement was started at an obscuration of ~10 %, followed by sonication (60 s, 70 %). The grain-size distribution was calculated following international standards (ISO 13320: 2020). The particle size statistics were calculated with the software GRADISTAT (Blott and Pye, 2001). The fraction

of median grain size (Q50) and the coarser 90[th] percentile (Q90) parameters within graded beds were used to further characterize event deposits (Wilhelm et al., 2013). Heatmaps of closely spaced grain-size data were calculated using the kriging method in Surfer 11.

For organic geochemistry, the sediment cores were described and afterwards sampled in 1 cm resolution. After lyophilisation, 1–2 cm³ of each sediment sample were homogenized with an agate mortar. $3.0 \pm 0.3$ mg of each sample was weighed into silver capsules for decalcification on a hot plate (70° C) using 5 % HCl until no effervescence was observed. After complete drying, capsules were folded and total organic matter content (TOC, wt%) and carbon isotope composition of bulk organic matter ($\delta^{13}C_{TOC}$) were determined using an elemental analyser (NC 2,500, Carlo Erba, Italy) coupled to an isotope-ratio mass

spectrometer (DeltaPlus, Thermo-Finnigan, USA). Elemental standards Atropine and Cyclohexanone 2,4-dinitrophenylhydrazone were used for calibration of carbon content, and IAEA-CH-7 and USGS41 for isotope calibration. Additionally, a lab standard (peptone) was used for linearity correction and isotope calibration. All isotope values are reported in the common $\delta$-notation.

Event deposits were dated using the previously published age-depth model (Oswald et al., 2021; Fig. S2) established by

Bayesian age–depth modelling of radiocarbon ages using the R-software Bacon v 2.4.3 (Blaauw and Christen, 2011) combined with ages derived from the peak fallouts in 1986 CE and 1963 CE of the radionuclide $^{137}Cs$ and by constant flux–constant sedimentation rate (CFCS) modelling of excess $^{210}Pb$ activities using the R-package SERAC (Bruel and Sabatier, 2020). Radiocarbon ages were calibrated with IntCal20 (Reimer et al., 2020) and reported in years before the common era (BCE) or common era (CE).

For the age–depth modelling, event deposits >5 mm were removed to obtain an event-free sediment depth. Radiocarbon samples are derived from organic macro remains in finely laminated intervals of background sediment (Table S2).

### 3.3.3 Sedimentary event identification

Debris flows form concentrated density flows in a lake which deposit turbidites with distinct sedimentological characteristics at the lake bottom, such as color, texture, grain-size, and organic content (Sletten et al., 2003; Irmler et al., 2006). Df turbidites

are sharp-bounded units with a fining-upward grain-size trend, often bearing terrestrial macro-remains (Sletten et al., 2003). In contrast, strong earthquake shaking triggers multiple subaquatic slumps that evolve into turbidity currents and generate a turbidite in the depocentre (e.g Schnellmann et al., 2002), hereafter referred to as "eq turbidite", with distinct sedimentological characteristics, such as a homogeneous grain-size trend and an in-lake geochemical fingerprint. In Plansee, previously interpreted eq turbidites are 5 to 35 cm thick amalgamated turbidites indicating deposition of multiple turbidity currents within

a short time (Oswald et al., 2021). All other event deposits were not the focus of detailed analyses of that study. Given the setting of the main basin with no river inflow, but surrounded by numerous debris-flow fans (Sect. 2), it can be hypothesized that these event deposits almost exclusively are debris-flow related. Thus, this study presumes this two-fold classification of lacustrine event deposits and tests this hypothesis by combining the amphibious geomorphological analyses, turbidite thickness distribution and sediment core analyses.

### 3.3.4 Spatial and temporal analyses of df turbidites

Thickness measurements of individual events in several cores provide means to visualize the spatial extent and geometry of the respective event deposit (Moernaut et al., 2014), which offers insights for sediment dynamics or source of the turbidity current. Therefore, we calculated the percent thickness distribution for each individual event by its thickness in a core relative to the accumulated thickness of the deposit in all four cores. Moreover, deposit thickness of flood-induced hyperpycnal flow

deposits have been calibrated in several alpine environments to represent flood intensity (e.g. Czymzik et al., 2013; Wilhelm et al., 2017); we also aim to test the applicability of this approach for df turbidites.

Phases with similar frequency of df turbidites (df phases) were statistically distinguished by a change-point analysis of the inter-event periods using the R function *cpt.meanvar* with a clustering algorithm *BinSeg* following Albrecher et al. (2019) and references therein.

Additionally, we calculated the annual occurrence rate of df turbidites using a central running sum with different bandwidths to reconstruct changes in debris-flow frequency over time in different resolutions. First, a suitable bandwidth (150 yrs) was selected based on the average df turbidite occurrence over the entire core using the mean age-depth model (Sheather and Jones, 1991). We applied this bandwidth to the occurrence rate calculation for each individual simulation of the age-depth model derived from the R-software Bacon v 2.4.3 (Blaauw and Christen, 2011) and calculated the quantiles of the yearly df turbidite

occurrence rate (0.025, 0.5 and 0.975) to obtain a 95% frequency probability. The actual window size for the running sum is continuously reduced at both edges once the sample is smaller than half of the selected bandwidth. This results in a data-based frequency analysis that incorporates age-depth model uncertainties. The rather broad bandwidth is suited for showing general changes in frequency over time. To account for a higher resolution in frequency changes especially in periods with higher number of events, we also calculated a bandwidth based on the df occurrence of the last two centuries and applied the resulting

21-yr bandwidth to occurrence rate calculation of the main age and 95% frequency probability of all individual age-depth model simulations. The cumulative thickness over time involves both the thickness and frequency of df turbidites and its slope provides information on df turbidite accumulation rate per year. We calculated the cumulative thickness on the mean values and the 95 % range values of the age-depth model to transfer age uncertainty to the cumulative thickness analysis.

## 4    Results

### 4.1    On land and underwater characterization of the alluvial fans and fan deltas

The investigated alluvial fan deltas form conical to lobate sediment accumulations protruding into the lake in front of funnel- to cirque-shaped catchments (Fig. 1). Onshore, the fan deltas can be subdivided into an active part and a partially active part dependent on geomorphic characteristics and vegetation type (Fig. 1e–f and 2). The active part is characterized by braided lobes and channels, cutting into older fan deposits, and is mostly vegetation free (Fig. 2a, d). In contrast, partially active parts

have clearly smoother topography due to gravitational hillslope processes and are abundantly occupied by shrub and tree vegetation including pines. There, debris-flow activity is indicated by small lobes and channels close to the active channel levee representing sediment spill-overs during large debris-flow events exceeding the channel capability. The onshore mean slope angle of the alluvial fan deltas varies from 10° for the lobate, mature fan delta to 24° for the cone-shaped, juvenile fan delta. Independent of these parameters and maturity of the fan itself, the mean slope angle shifts to ~30° once a fan submerges

into standing water.

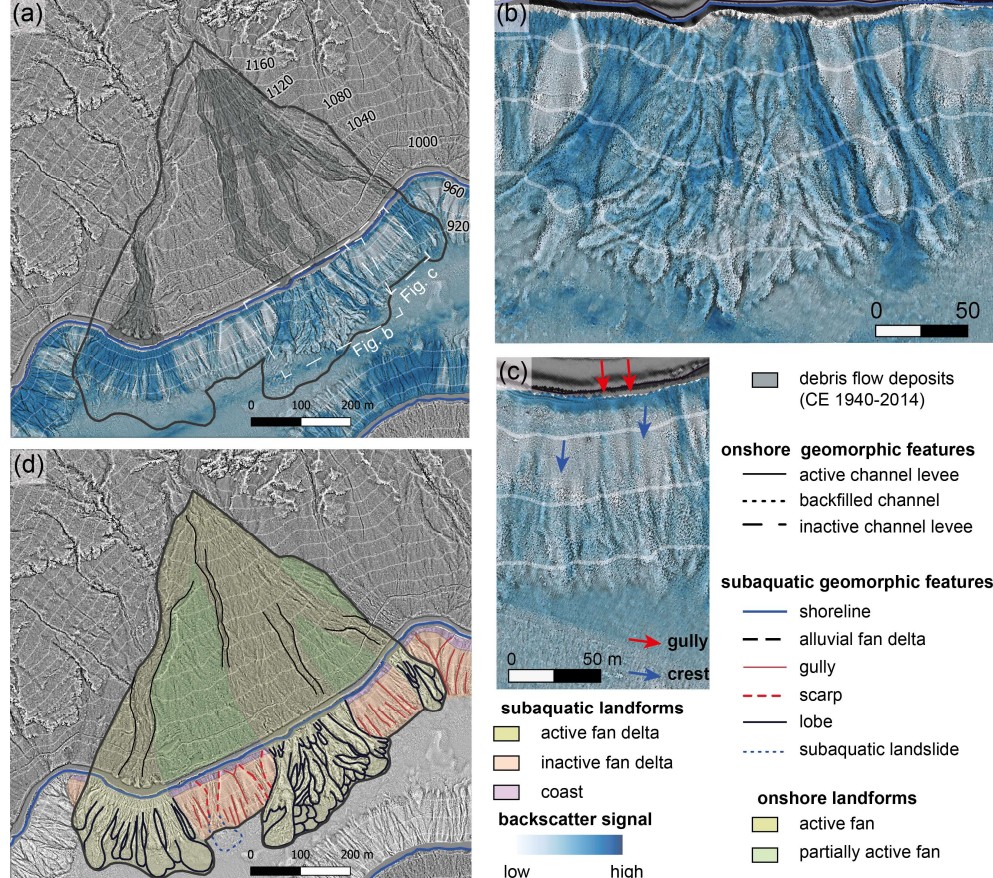

**Fig. 2: Combined onshore and offshore geomorphic characterization of the calibrate fan delta with known recent debris-flow activity. a) Topographic openness difference map overlain by the multi-beam backscatter data (blue colorbar) and the documented debris-flow deposits between 1947 and 2014 (Dietrich and Krautblatter, 2017). b) Zoom of an active part of the fan delta showing stacked and braided debris-flow lobes with intermediate to high backscatter signals. c) Zoom of an inactive part of the fan delta showing the relatively regular pattern of crests and gullies. d) Geomorphic-interpreted fan delta showing characteristic terrestrial and subaquatic landforms for active and inactive fans and fan deltas. Onshore DEM derived from Land Tirol (data.tirol.gv.at).**

The subaquatic geomorphology of alluvial fan deltas has a greater variety in geomorphic features than its onshore part (Fig. 2a, d). In general, three subaquatic landforms are subdivided: the coastal zone, an active fan delta, and an inactive fan delta. The coastal zone is characterized by a 5–10 m broad area of a high backscatter signal indicating abundant coarse (sand-gravel) sediments (Fig. 2a, d). A coarse-grained coastline occurs almost everywhere independent whether the part of the fan is active or inactive. The subaquatic active fan delta is generally characterized by a bulge in respect to the general arcuate trend of the fan (Fig. 2a, b, d). In the detailed view of Fig. 2b, d, the bulge consists of numerous stacked and braided debris-flow lobes. Some of the lobes have intermediate to high backscatter signals indicating coarse-grained and thus recent debrites, whereas older debrites are expected to be covered in fine-grained lacustrine mud and thus exhibit low backscatter values. This observation is in accordance with detailed mapping of onshore debris-flow deposits which have recently occurred between 1947 and 2014 CE (Dietrich and Krautblatter, 2017), as the subaquatic continuations of these mapped debris-flow deposits exhibit high backscatter values (Fig. 2a). In contrast, the inactive fan delta is characterized by a more regular subaquatic morphology dominated by parallel downslope-oriented gullies and crests (Fig. 2a, c, d). The backscatter signal is low at the gentle slopes representing inactivity of coarse detrital sedimentation and already sufficient coverage with fine-grained lacustrine mud. Intermediate backscatter signals occur along gullies and at the basin near the slope-break (Fig. 2c).

At the latter location, coarse sediments are deposited by debris-flow activity or remobilization and transportation of coarse coastal material mixing with fine-grained lacustrine basin floor sediments. The cause of abundant coarse material in gullies is interpreted to be coastal erosion and related density flows which either deposit coarse coastal material or erode a possible fine-grained lacustrine sediment cover and can thus expose old, coarse debrites. Independent of its underlying process, the more or less regular pattern of gullies and crests seems to develop at the inactive fan delta due to subaquatic gravitational slope

processes, which cannibalize the previously deposited debris-flow lobes and thus fundamentally alter the subaquatic geomorphology created by debris-flow processes. Funnel-shaped landslide scars also occur at inactive fan delta slopes and locally the corresponding subaquatic landslide deposit can still be observed in bathymetric data (red and blue dashed lines in Fig. 2d).

**Fig. 3: Topographic openness difference maps of the juvenile (a) and mature (b) fan deltas (left) and their geomorphic interpretation (right). The steep and small juvenile fan (a) has a channel backfilled with sediment, a currently inactive channel and a distinct active channel cut into previous fan deposits. The active subaquatic fan delta shows numerous stacked debris-flow lobes and a subaquatic delta failure in form of a subaquatic landslide. The large and flat mature fan (b) shows beside a distinct active channel several diffuse inactive channels. Subaquatic debris flow-induced turbidites (df turbidites) along diffuse inactive channels are interpreted as debris-flow events spilling over the currently active channel levee (orange arrow). The active fan delta is composed of only few df turbidites at the basal slope in the elongation of gullies interpreted as extraordinary large debris-flow events (blue arrows). A detailed legend is provided in Fig. 2. Onshore DEM derived from Land Tirol (data.tirol.gv.at).**

The two fan deltas which were the focus of the repeated TLS investigations (see Sect. 4.2) represent two geomorphic end members—mature and juvenile—and their landforms and characteristics are hereafter described in more detail.

The western juvenile fan delta is small (0.11 km²), steep (24° on average) and has a conical shaped apron with a dominant, deeply incised active channel eroding into previous fan deposits (Fig. 3a). The apron contains an inactive to partly active channel and a backfilled channel (dashed and dotted line in Fig. 3a, respectively). The fan catchment (0.18 km²) is funnel-shaped (Fig. 1a). In the subaquatic realm, the fan has a slightly arcuate shape due to its prominent active fan delta in the middle of two inactive areas. Numerous stacked debris-flow lobes built up the active part indicating active fan progradation. A small delta-failure event occurred in the active part and is indicated by a subaquatic landslide deposit and its corresponding scar (Fig. 3a). In the basin near the basin-slope transition an irregular, hummocky morphology shows relics of older and larger fan delta failures buried by lacustrine mud.

The eastern mature fan delta is much larger (0.36 km²) and depicts a lobate-shaped fan with a low mean slope angle (10°). The large catchment (1.19 km²) is bowl-shaped and extends over several side valleys (Fig. 1a). The fan delta has one distinct active channel in the centre and several smaller diffuse currently inactive channels (Fig. 3b). On a topographic profile along the main flow axis, the onshore part of the fan has a concave shape with a trend to lower slope angles towards the lake. The subaquatic morphology is characterized by an overall arcuate apron shape at the slope–basin transition. A few smaller bulges against the general trend represent the current or rather recent active fan part prograding in the lake. The inactive fan part is dominated by parallel crests and gullies and shows a few recent debris-flow lobes, which contrasts with the lack of these lobes for inactive parts of the steeper fans e.g. the juvenile fan (Fig. 3a). These lobes often match with the subaerial diffuse channels and thus might represent deposits of the events spilling over the currently active levee (orange arrows in Fig. 3b). Additionally, the inactive fan part contains a few subaquatic landslide scars (red dashed lines in Fig. 3b). The currently active fan part has only a few stacked debris-flow lobes in the upper part of the subaquatic slope, while most of the lobes occur in the lower slope in front of an incised gully (blue arrows in Fig. 3b). This implies that gully formation partially overprinted the debris-flow lobes and potentially hints that, in recent times, most sediment gets accumulated onshore on the shallow fan and only during extraordinary large events a significant debris-flow volume reaches the subaquatic slope.

In comparison, the juvenile fan is characterized by a smaller fan area, a higher average slope angle and a smaller fan–catchment size ratio than the mature fan. On the mature fan, talus is removed by a small perennial stream in addition to episodical sediment transport. The terrestrial profile of the mature fan extends into the lake, creating a low inclination depositional area, which contrasts the convex shape of the juvenile fan. The latter shows active progradation in the subaquatic realm, whereas the mature fan displays less signs of recent debris-flow activity.

## 4.2 TLS-measured net topography change of alluvial fans

The net topography changes of the onshore fan delta surfaces range between -2.6 m and +1.8 m from May to August 2019. The total mobilized volume during the 3-month period is 1.9 times higher on the steep, juvenile fan delta compared to the flat, mature fan delta. This difference is potentially linked to the catchment topography and connectivity (i.e. sediment throughput) of the individual fan delta.

The steep, juvenile fan delta displays a maximum of 1.6 m increase in height and a maximum erosion depth of 2.6 m (Fig. 4). Erosion and deposition are balanced for the confined fan delta and the channel geometry was drastically altered during the investigated period. The fan topography on May 10th displayed a shallow, continuous channel which has eroded previous deposits on the fan apron and was partially backfilled in the lower third of its extent. Since most subaquatic depositional lobes are found in elongation of this channel (see Sect. 4.1), it is assumed to be the dominant pathway for channelized flows. The second survey on August 22nd reveals backfilling and overtopping of the former channel (Fig. 4). There are two zones of deposition in the lower fan area: a funnel-shaped debris accumulation in the terminal part bordering the hiking trail and a widespread, more proximal zone of deposition, which represents progressively short and wide debris flows. The erosional zone extends from a small ravine to a deeply incised U-shaped channel cutting the talus above the newly developed zone of deposition.

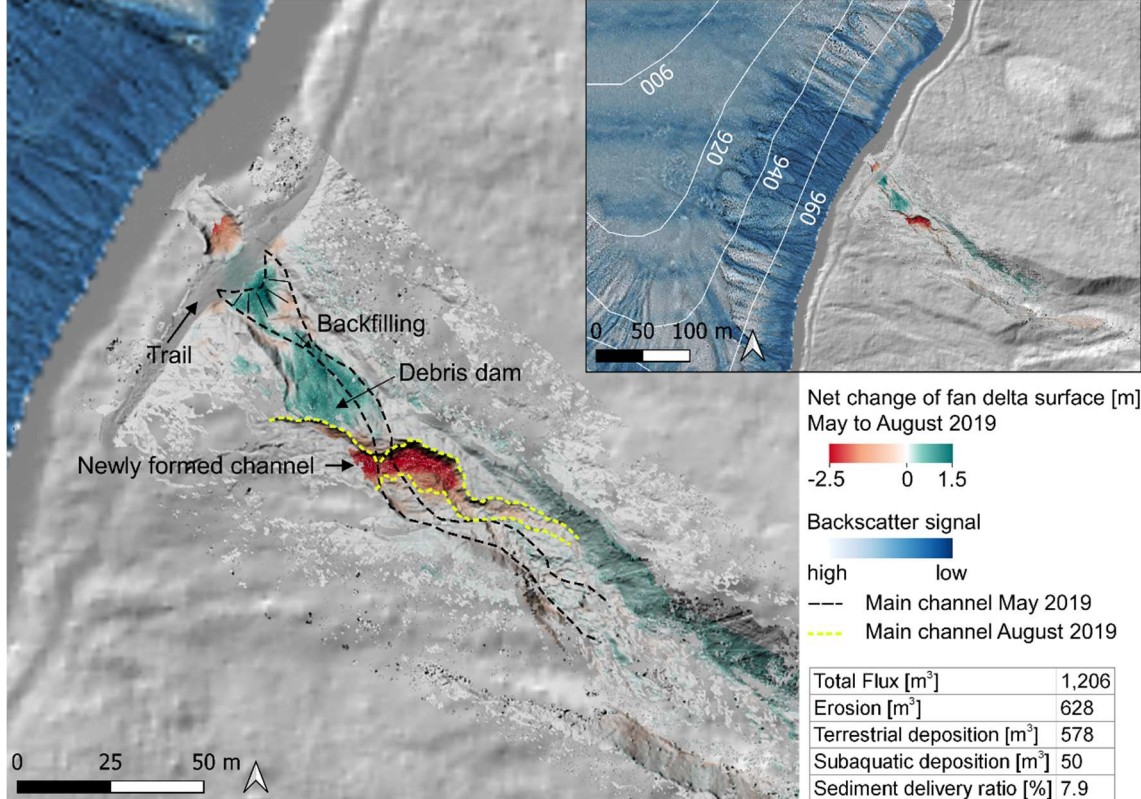

Fig. 4: Net topography change of the steep, juvenile fan delta with 0.18 km² catchment between May 10th and Aug 22nd 2019, calculated from surface comparison of TLS data. Total flux represents total eroded and deposited volume on LiDAR covered fan delta. The erosion volume exceeds the deposition volume, leading to sediment delivery into the lake. The August survey reveals backfilling and overtopping of the former channel. The formation of a debris dam obstructed subsequent debris-flow surges and led to a new channel incision. This fan type displays interfingering channels, an unstable morphology and high sediment flux over the investigated period. The juvenile fan delta is dominant at Plansee, therefore most debris-flow events on the surrounding slopes form subaquatic deposits on the lake floor. Onshore DEM derived from Land Tirol (data.tirol.gv.at).

Stratified deposits of alternating coarser and finer layers representing interfingering channels were exposed on the scarps, and the new channel displays abundant coarse clasts. A second zone of erosion formed below the trail, indicating sediment transport into the lake. Over the whole debris-flow track, the eroded volume (628 m³) exceeds the deposited volume (578 m³), resulting in a 50 m³ sediment flux into the lake and a sediment delivery ratio of 7.9 % for the investigated period.

During the 3-month period, onshore sediment deposition dominated the shallow, mature fan delta (Fig. 5). The proximal fan area shows a parallel shift in the debris-flow track between May and August 2019. On the distal fan area, the new flow track is connected to the former main channel, where sediment deposition is concentrated with increasing height towards the subaerial–subaquatic transition (up to 1.8 m). The rest of the active fan surface experienced an elevation increase of up to 0.8 m. Erosion occurred accessorily on the channel bank and parallel to the shoreline, forming narrow linear structures of up to 1.2 m height loss. Large clasts were spread over the entire cross-section of the terrestrial fan delta. In total, 41 m³ of debris were eroded and 651 m³ were deposited onshore, while no sediment flux into the lake was detected in the investigated period.

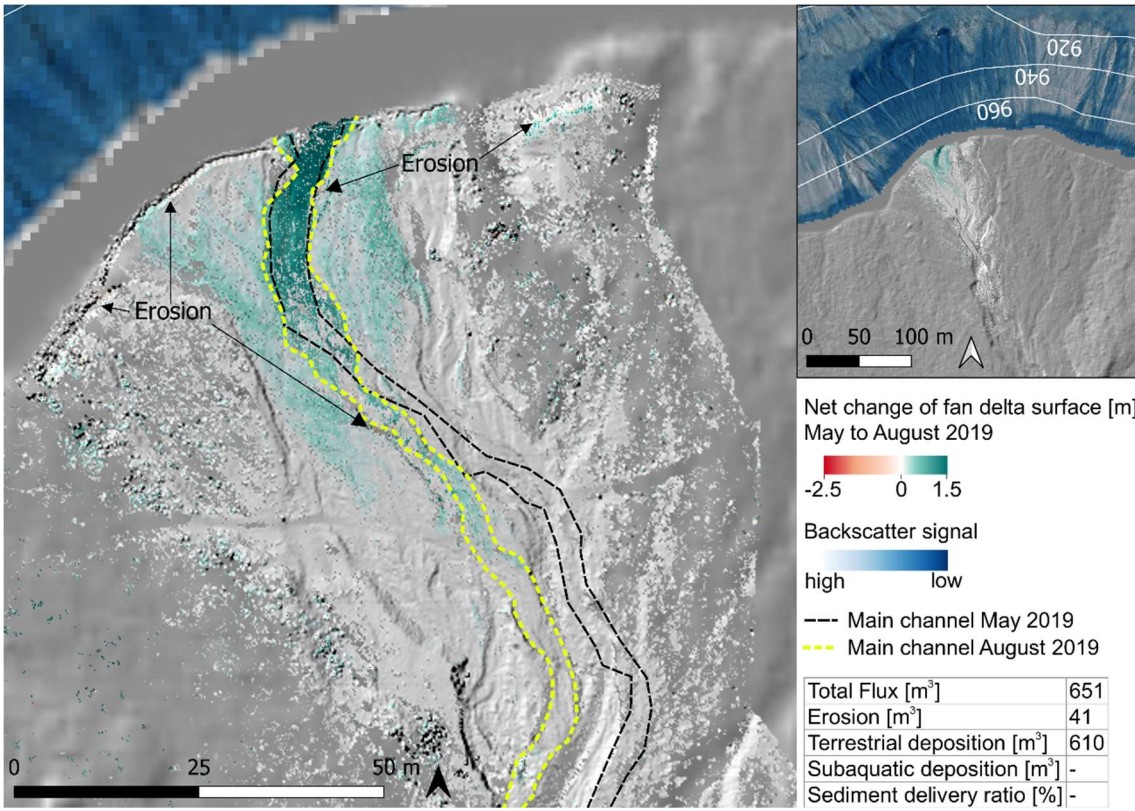

**Fig. 5: Net topography change of the shallow, mature fan delta with 1.19 km² catchment between May 10th and Aug 22nd 2019 calculated from surface comparison of TLS data. The total sediment flux (i.e. total eroded and deposited volume on LiDAR covered fan delta) is lower compared to the juvenile fan delta of Fig. 4. Deposition dominates and linear erosion occurs accessorily on the channel bank and parallel to the shoreline. The mature fan delta is dominated by terrestrial deposition and no sediment flux into the lake was detected. The main channel and levees were preserved in the lower fan area. This fan type shows a stable morphology and widespread onshore deposition. Onshore DEM derived from Land Tirol (data.tirol.gv.at).**

## 4.3 Lacustrine event deposits

### 4.3.1 Event type differentiation

Lacustrine sedimentation in the main basin of Plansee is generally characterized by dark-grey to ochre, finely laminated clayey silts with abundant detrital carbonates and subordinate contents of diatoms and organic matter (background sediment; Fig. 6a). The background sediment contains 1.5–2.3 wt% total organic carbon (TOC) with C/N ratios between 13 and 19 and $\delta^{13}$C values between -28.0 and -30.4 ‰ (Fig. 6b; Table S3). This indicates a mixture between lacustrine organic matter of algal origin (typically with C/N ratios <10) and terrestrial organic matter with $TOC/TN_{molar}$ ratios typically exceeding a value of 20 (Meyers and Teranes, 2001). Two different types of event deposits are intercalated in the background sediment, which can be macroscopically and analytically distinguished (Fig. 6a-c).

One event type consists of a grey, homogenous coarse silt turbidite with a thin fining upward base as e.g. present at core depth 116 cm in Fig. 6a. This event type, here referred to as "eq turbidite", was related to events of multiple subaqueous mass-wasting caused by strong seismic shaking (Oswald et al., 2021) and occurs only at three stratigraphic levels in the short cores (Fig. 7a), corresponding to earthquakes at ~2120 BCE, ~1050 BCE, and 1930 CE (Namlos M 5.3 earthquake). Eq turbidites yield around 0.7–2.3 wt% TOC and $\delta^{13}$C values between -26.9 and -29.2 ‰ (Fig. 6b; Table S3). Relatively constant lightness L* and density values support the homogeneous character of eq turbidites except the deposits of eq turbidite 3 in Plan 19-03 and Plan 19-04 where a subaqueous landslide deposit characterized by contorted strata overlain by a graded turbidite was cored, and which corresponds to a strong earthquake at ~2120 BCE (Oswald et al., 2021; eq3 in Fig. 7a).

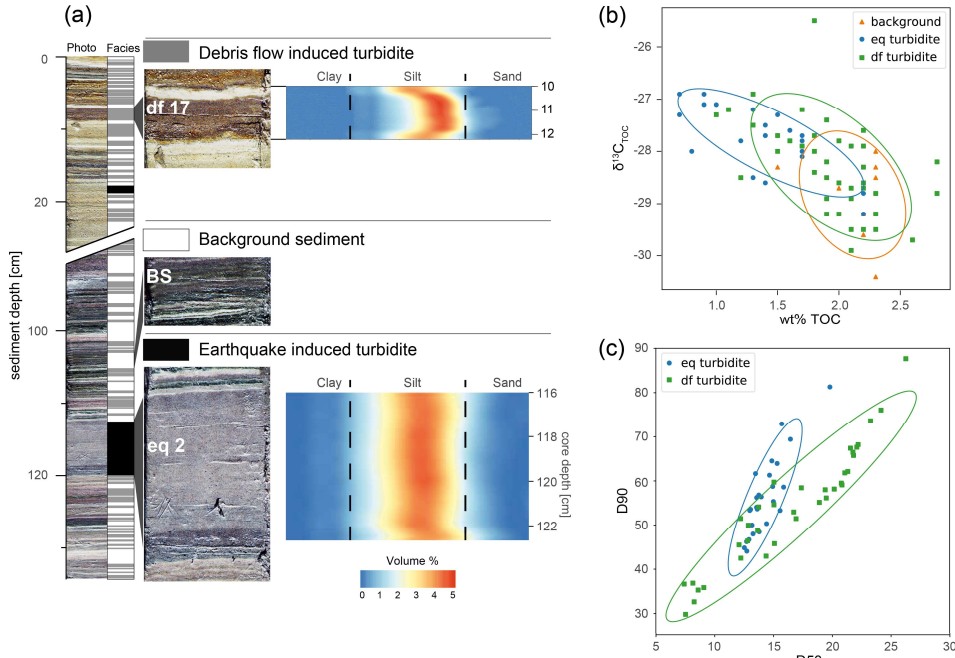

Fig. 6: Event deposit characterization in sediment core Plan 19-02. a) Core image of Plan 19-02 and corresponding core log shows finely laminated background sediment intercalated by debris flow-induced turbidites (df turbidites) and earthquake-induced turbidites (eq turbidites). Df turbidites have a gradual fining-upward grain-size trend, whereas eq turbidites are homogeneous deposits on top of a thin coarse base. Eq turbidites show broader grain-size distributions (poorer sorting). b) TOC/δ¹³C plot of samples from all cores shows general differences for df- and eq turbidites and similarities between df turbidites and the (predominantly clastic) background sediment. c) D50/D90 grain-size plot of df- and eq turbidite samples from all cores highlights the different evolution of df- and eq turbidites. Colored ellipses represent 95 % confidence ellipse.

The second event type (df turbidite) is characterized by brown to ochre colored, up to 6.5 cm thick detrital deposits, which abundantly occur throughout the sediment cores (Fig. 6a). Df turbidites generally have a sharp or irregular coarse-grained base (coarse silt to fine sand) overlain by a progressive fining upward sequence and a fine-grained (fine silt) top. Some of the df turbidites have a coarsening upward trend at the base with the maximum grain size in the lower-middle part followed by a normal grading (Fig. 6a). Such grain-size trends in detrital deposits indicate waxing and subsequent waning of flow-energy transporting the terrestrial sediments into the lake during a single high discharge event (e.g. Gilli et al., 2013). Terrestrial organic macro-remains often occur bedding-parallel aligned at the base or in the middle part of the deposits. TOC values ranging from 1.0 to 2.8 wt% are higher compared to eq turbidites (Fig. 6b; Table S3). The lower TOC contents in eq turbidites are potentially caused by the decomposition of organic matter on subaquatic slopes prior to the earthquake-induced remobilization of the slope deposits, whereas higher TOC contents in df turbidites show no sign of decomposed organic matter (Vandekerkhove et al., 2020). On average, df turbidites also show slightly lower δ¹³C values from -25.5 to -29.9 ‰. In addition, the grain-size evolution patterns of df- and eq turbidites derived from a D50 versus D90 diagram are different (Fig. 6c). Eq turbidites have a steeper trend, suggesting a poorly sorted turbidite caused by subaqueous mass movements (Wilhelm, 2012). In contrast, df turbidites mainly follow a much less steep trend (Fig. 6c), which is commonly related to well-sorted density flow deposits induced by flood events (Wilhelm et al., 2017).

All above mentioned observations and characteristics of these event deposits suggest the interpretation that df turbidites evolved from terrestrial debris flows. The potential of misinterpreting df turbidites as river flood-induced turbidites, which could have similar characteristics (Gilli et al., 2013; Wilhelm et al., 2013), is very low at this subbasin of Plansee. This is because possible hyperpycnal flows related to the main inflowing rivers are trapped either in Heiterwangersee or in the easternmost subbasin in Plansee and do not reach the studied main basin (Fig. 1a).

In addition to the event-type distinction based on sedimentological and geochemical parameters, identification and mapping of the respective geomorphic expressions in bathymetric data (Fig. 3a) allows conclusive interpretation on df- and eq turbidites.

#### 4.3.2 Spatio-temporal distribution of df turbidites

Sedimentation processes and temporal occurrence of df turbidites offshore the juvenile fan delta were investigated in four short cores with 50–75 m spacing, forming a proximal-to-distal transect from the juvenile fan delta towards the main basin depocentre (Fig. S1). The two most distal cores (Plan 18-10 and Plan 19-02) are located in the flat depocentre at 77 m water depth with low slope angles (<1°), whereas the more proximal cores Plan 19-03 (76 m water depth, 2° slope angle) and Plan 19-04 (75 m water depth, 3° slope angle) are located on the gentle slope towards the juvenile fan. The most proximal core Plan 19-04 is located close to the slope break. The short cores share the overall lithostratigraphic succession composed of four lithotypes (LT1-LT4), which allows for cross-correlation of the 138 identified event deposits (Fig. 7a, Table S4). Additionally, combined radiocarbon and short-lived radionuclide dating of the most distal core Plan 18-10 provides age information on the stratigraphic succession and on the temporal distribution of df turbidites (Figs. 7a, S2, Table S2).

LT1 (~2120 to ~2040 BCE) consists of predominantly df turbidites (in total 10) with hardly any background sediment in between the df turbidites and is intercalated by the eq3 turbidite (~2120 BCE). LT2 (~2040 BCE to ~1520 CE) is characterized by brown- to ochre colored background sediment with relatively regularly intercalated df turbidites (in total 80) and the eq2 turbidite at ~1050 BCE. LT3 (~1520 to ~1920 CE) contains light-grey to ochre background sediments with intercalation of 17 df turbidites. The uppermost part of LT4 is characterized by a grey sedimentary facies with abundant deposition of 31 df turbidites and contains the eq1 turbidite at 1930 CE. Dark-grey sediments at 3–12 cm sediment depth in LT4 are likely related to eutrophication in the mid to late 20th century (Schindler, 2006).

A 2D spatial distribution of df turbidites is obtained by the deposit thickness distribution in all four cores (Fig. 7b), which provides potential information on sedimentation processes and source areas. Most df turbidites are relatively thicker in the two distal cores compared to the two more proximal cores (Fig. 7b), indicating a ponding geometry of the turbidite body resulting from sediment-laden density flows, i.e. underflows (Gilli et al., 2013). Several df turbidites highlight this ponding-deposition character by more than 70 % thickness distribution in the two distal cores (grey arrows Fig. 7b). Extreme examples for pronounced distal sediment deposition are present in sequences in LT4 (df 11–16) and at the end of LT1 (df 128–129) where the events exclusively occur in the distal cores. In contrast, several df turbidites are thicker in the proximal cores and occur throughout the sequence (black arrows Fig. 7b). An exclusive deposition in proximal cores only occurs in LT1 (df 130–138; Fig. 7b), for which no df turbidites are present in the distal core Plan 18-10.

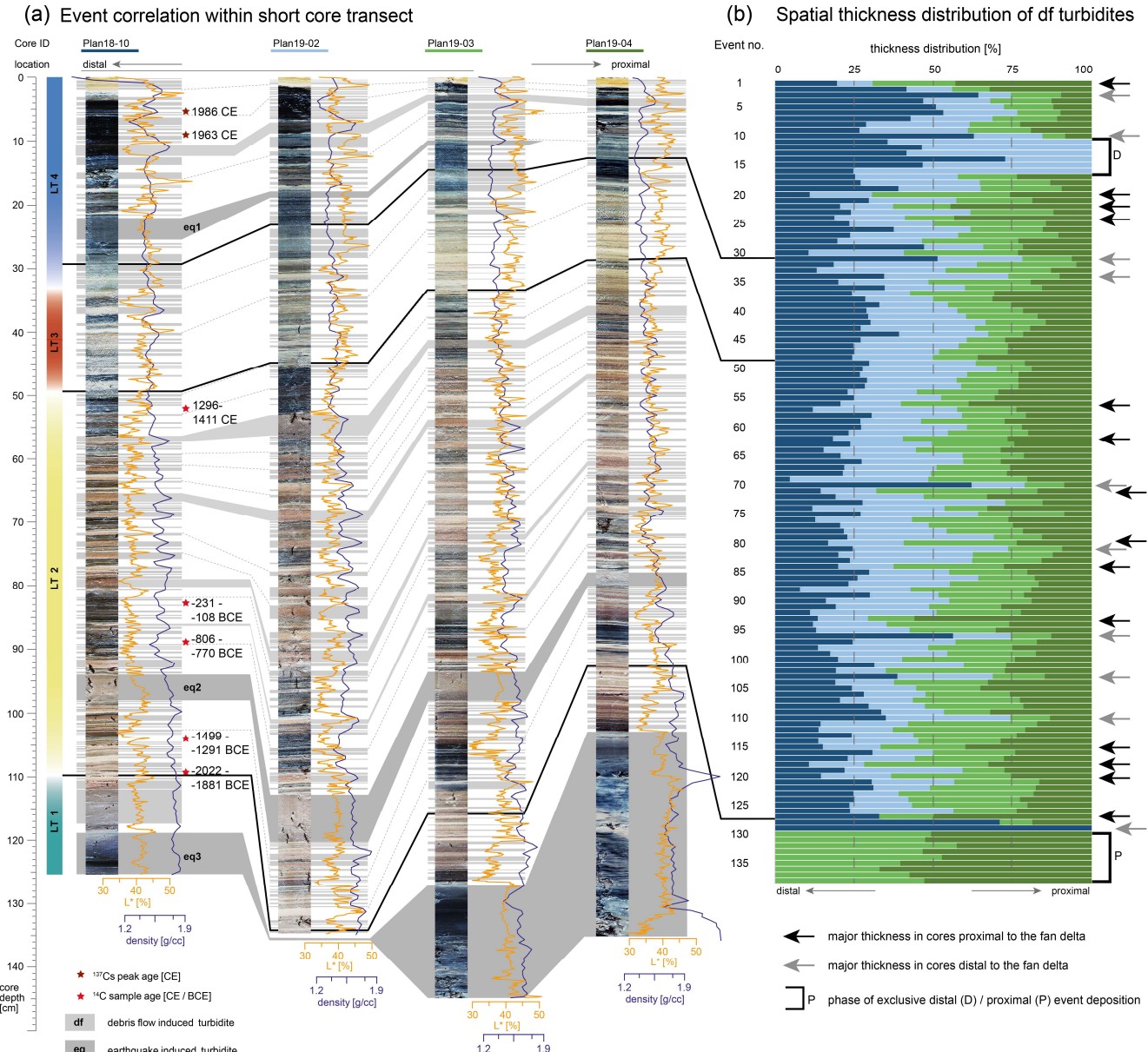

**(a)** Event correlation within short core transect

**(b)** Spatial thickness distribution of df turbidites

**Fig. 7: Core-to-core correlation of distal (left) to proximal (right) short cores in respect to the juvenile fan delta. Four lithostratigraphic types can be distinguished based on the initial core images, bulk density (violet) and lightness (orange) and dated by combined radiocarbon (red stars) and short-lived radionuclide ($^{210}$Pb and $^{137}$Cs; dark-red stars) dating (Fig. S2). Each of the 138 identified debris flow-induced (df) and 3 earthquake-induced (eq) turbidites are cross correlated in the four cores and measured for their thickness. b) Thickness distributions of each df turbidite in the four short cores (individual thickness relative to accumulated thickness in the four cores) are color-coded based on the corresponding core and show different deposition patterns for different events and phases.**

For further analyses on past debris-flow frequency and intensity, the core Plan 18-10 is considered as the most representative for an undisturbed sequence due to its location in the depocentre where the potential erosive power of underflows is lowest and because it contains the most complete event stratigraphy deduced from core correlation. This location is also considered to potentially best represent event intensity derived from thickness measurements due to the overall ponding geometry of df turbidites. The few missing events (df 130–138) are projected to the master core Plan 18-10 to guarantee record completeness for further analyses. Four main phases with similar frequency of df turbidites (df phases) are differentiated by a change-point analysis of the 4,000 year debris-flow record (Fig. 8a, Tables 1, S2). Strikingly, the autonomously differentiated df phases using a change point analysis on the inter-event times of df turbidites temporally coincide with the above-described lithotypes indicating the major influence of debris flows on subaquatic event deposition in Plansee.

The frequency analysis of the 4,000 year df turbidite record is calculated on a 150-yr and a 21-yr bandwidth. These bandwidths are based on a bandwidth selection test of the event number and number of age constraints and are thus only valid for certain periods. To account for higher resolution changes especially in periods with higher number of events, we also calculated a

bandwidth based on the df occurrence of the last two centuries and applied the resulting 21-yr bandwidth to occurrence rate calculation of the main age and 95% frequency probability of all individual age-depth model simulations.

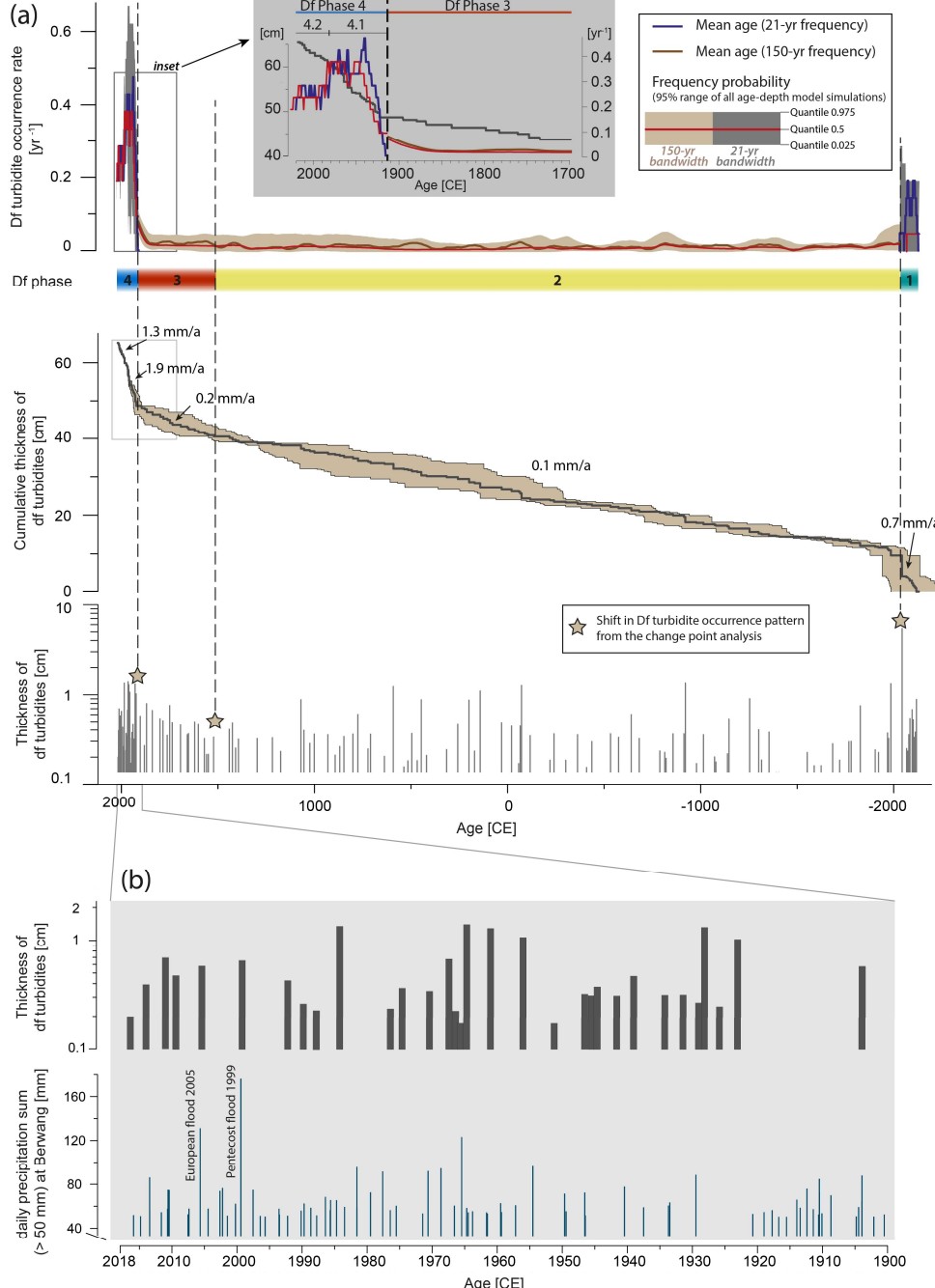

510

**Fig. 8: Temporal distribution of debris flow-induced (df) turbidites in the distal core Plan 18-10. a) Temporal- and thickness distribution of the 138 df turbidites is displayed based on their mean age (bottom). The cumulative thickness of the df turbidites over time given the mean event ages (black line) and their modelled 95 % probability range (light-brown area) are shown in the central panel. In the top panel, the annual occurrence rate of df turbidites is displayed as the annual frequency of the mean age (21-**
515 **yr and 150-yr bandwidth) and the 95% frequency probability bracketed by the 0.025 and 0.975 quantiles of all 6396 individual age-depth model simulations (150-yr bandwidth: light-brown; 21-yr bandwidth: grey). Note that the 21-yr and 150-yr bandwidth frequencies are only valid for certain periods defined by a bandwidth selection test and are over-/undersampled for other periods. A comparison of the frequency probabilities from the two bandwidths is provided in Fig. S3. The four df phases (phases of similar debris-flow activity derived from a change-point analysis of interevent times) are delimited by vertical dashed lines across these**
520 **three plots. The grey-shaded inset in the top panel shows a zoom of the cumulative thickness and 21-yr and 150-yr frequency of debris flows in the period 1700–2018 CE. Note the good fit of the yearly df turbidite frequency from the 0.5 quantile with the mean age from the 21-yr bandwidth in the last century. Phase 4 is subdivided based on the frequency changes into a high-frequency phase 4.1 and a phase 4.2 with lowered, but still 7 times higher frequency than phase 3. b) Comparison of df turbidite thickness since 1900 CE with daily precipitation sums (>50 mm) at the 7 km distant weather station Berwang (Hydrographischer Dienst Tirol, 2020)**
525 **highlighting the potential temporal coincidence of the strongest flood events in 1999 and 2005 CE with df turbidites. Mean age uncertainties are ± 6 years (2018–1960 CE) and ± 19 years (1959–1920 CE; see also Table S4). Note that rainfall intensity and thickness distribution do not correlate and not every year with high daily precipitation (>50 mm) has a corresponding df turbidite and vice versa.**

The temporal distribution and thickness of df turbidites (Fig. 8a) in combination with derived statistics (recurrence interval, mean thickness, and deposition rate; Table 1) strongly vary in the main df phases 1–4. The oldest phase 1 is characterized by abundant df turbidite occurrence represented by relatively higher event frequencies with a mean recurrence interval of 7.4 years (Fig. 8a; Table 1) and by containing the thickest event deposits of the whole record. High event frequency and thick event deposition is also reflected by the steep cumulative thickness trend with a debris-flow deposition rate of 0.7 mm a$^{-1}$. Df phase 2 is characterized by sporadic df turbidites, centuries of event quiescence, the largest recurrence interval of 43.9 years, and the lowest debris-flow deposition rate (0.1 mm a$^{-1}$) of the sedimentary record. The few events in this phase have average thickness reflected by vertical steps in the cumulative thickness plot (Fig. 8a), indicating that low event frequency does not imply smaller event thicknesses. Phase 3 is represented by slightly increased event frequency reflected by e.g. the decreased mean recurrence interval of 20.9 years and an increased debris-flow deposition rate of 0.2 mm a$^{-1}$ compared to phase 2. The youngest phase 4 exhibits the highest abundance of df turbidites with a mean interval of 3.0 years (Table 1), which is also reflected by the steepest slope in cumulative event thickness with a mean debris-flow deposition rate of 1.7 mm a$^{-1}$. In detail, df phase 4 can be subdivided into two sub-phases based on human interferences in the second half of the 20$^{th}$ century likely decreasing df frequency (grey inlet in Fig. 8a; see also discussion point iv in Sect. 5.4). Df phase 4.1 is represented by a strong and fast frequency increase at ~1920 CE, followed by a period of highly frequent debris-flow events until ~1980 CE. Since then, the current df phase 4.2 has lower frequencies relative to phase 4.1 but still by far higher frequencies than in the main df phases 1–3. Debris-flow frequency in 4.1 increased by a factor of 8 compared to the reference df phase 3. In df phase 4.2, debris-flow frequency increased by a factor of 7 compared to df phase 3.

**Table 1: Lithostratigraphic phases and corresponding df turbidites including their recurrence interval, mean thickness and deposition rate. Uncertainties for recurrence interval and depositon rate are derived from the minimum and maximum ages derived from the age-depth model.**

| | | Df turbidites | | | | | | | | |
|---|---|---|---|---|---|---|---|---|---|---|
| Df Phase | Time (CE) | Total no. | Event no. | Recurrence interval (a/event) | | | Mean thickness (cm) | Deposition rate (mm/a) | | |
| | | | | mean | min | max | | mean | min | max |
| 4 | 2018 CE to ~1920 CE | 31 | 01-31 | 3.0 | 2.4 | 3.8 | 0.51 | 1.70 | 2.01 | 1.33 |
| *4.2* | *2018 CE to ~1980 CE* | *10* | *01-17* | *3.2* | *3.2* | *3.3* | *0.47* | *1.30* | *1.35* | *1.31* |
| *4.1* | *~1980 CE to ~1920 CE* | *21* | *18-31* | *2.5* | *1.6* | *3.7* | *0.55* | *1.90* | *2.48* | *1.07* |
| 3 | ~1920 CE to ~1520 CE | 17 | 32-48 | 20.9 | 12.9 | 26.4 | 0.46 | 0.20 | 0.31 | 0.18 |
| 2 | ~1520 CE to ~2040 BCE | 80 | 49-128 | 43.9 | 44.8 | 43.1 | 0.39 | 0.09 | 0.08 | 0.09 |
| 1 | ~2050 BCE to~ 2120 BCE | 10 | 129-138 | 7.4 | 4.4 | 11.3 | 0.96 | 0.70 | 1.78 | 0.64 |

The weather station in Berwang, 7 km SE of the coring site, has recorded daily precipitation sums since 1900 CE, which comprises most of phase 4 (Fig. 8b). The two most pronounced rainstorm events in 1999 and 2005 CE, corresponding to long lasting precipitation and river floods in several countries in central Europe (e.g. Barredo, 2007), potentially coincide with the timing of two df turbidites (df 5 and df 6). However, event thickness does not correlate with daily precipitation sum indicated by an only intermediate thickness of df 5 and df 6. In addition, not all df turbidites of the last century can be unambiguously linked to a measured rainstorm event, potentially due to age uncertainties, limits in macroscopic df turbidite detection, and in transport efficiency of debris flows (see discussion in Sect. 5.1 and 5.3).

# 5 Discussion

## 5.1 Data quality and justification of event characterization

Methodological limitations in our study principally relate to i) data quality of TLS and bathymetric data and TLS-based volume calculations, ii) first-order interpretation of the event characterization and iii) age-depth model uncertainty range.

i) Error sources in processing terrestrial laser scanning could derive from shrub vegetation coverage that was cut out from the point cloud and from the roughness of the terrain. Due to the poor vegetation cover in the debris-flow channel and multiple scanning positions, only minor deviations are expected for the volume calculation and the roughness was addressed by using

multiple complementary scan positions. The upper, inaccessible catchment parts could not be covered by LiDAR, but based on field observations we infer that they cannot accumulate significant amounts of debris and the sediment redistribution is concentrated in the LiDAR-covered channel and delta area. The lowest end of the newly formed channel on the juvenile fan delta was not covered entirely by LiDAR measurements, which might have influenced the erosion volume by a few percent, but TLS data certainly covered the main incisions. Erosion and deposition most likely occur during a few single events, but

consecutive minor redistribution could not be excluded.

In terms of bathymetric mapping, the uncertainty is derived from a combination of accuracy calculations for individual sensors (navigation, orientation, motion compensation), latency and sensor errors, and the application of sound velocity profiles (point observations) over the entire basin. Generally, the uncertainty is larger in areas with i) steep subaqueous terrain with irregular morphology resulting in low point density for these parts, ii) deep flat lake floor morphology with little data overlap from

independent survey stripes, and iii) where large beam angles are needed (central beams are more accurate than outer beams).

ii) The discrimination of the underlying process for different types of turbidites can be challenging especially in clastic influenced lake settings, where pure terrestrial input and remobilisation of subaqueous slopes only show subtle sedimentological and geochemical differences (e.g. Vandekerkhove et al., 2020). Yet, for the main basin of Plansee the initial hypothesis of a two-fold event categorization differentiating eq turbidites and df turbidites (outlined in Sect. 3.3.3) is supported

and justified by the data obtained from our amphibious geomorphological and sedimentological analyses. In this particular setting, the main basin has no permanent river inflow and is surrounded by numerous alluvial fans, which are dominated by debris-flow activity. A misinterpretation of df turbidites with flood-induced turbidites that would link to the permanent river inflows in the east can be excluded. River-flood induced turbidites are deposited in the 65 m deep main depocenter of the eastern basin, while the main basin is protected from the river deltas by circa 20 m high morphological barriers. Eq- and df-

turbidites have distinct and distinguishable sedimentological characteristics in Plansee especially on the basis of sediment color (L*), grainsize evolution, organic content and their distribution pattern.

Other potential mechanisms that could lead to a similar sedimentological signature as df turbidites are subaqueous fan delta failures initiated by excess pore pressure or sediment loading, which cannot fully be excluded but are unlikely at Plansee. The subaquatic slopes of the active fan deltas mainly consist of gravels and sands (see backscatter data of Figs. 2 and 3), the high

permeability of which does not allow excess pore pressure to develop. Some subaerial debris flows might not reach the subaqueous depocenter but add external sediment load on the fan delta which can subsequently initiate failure of the active fan delta slope. There might be a time delay between the subaquatic slope failure and the external sediment loading by the subaerial debris flow, or fan delta sediment remobilization may also be initiated by a subsequent subaerial debris flow event. In any case, df turbidites resulting from failure of the active fan delta slope are genetically linked to subaerial debris-flow activity.

Although they may not always correspond to an actual singular subaerial debris flow event in a one-to-one manner, they may represent delayed subaqueous fan delta slope failure. We potentially misinterpret some of the df turbidites that might be related to some other unknown trigger. However, a few wrongly interpreted turbidites will have little impact on the inferred debris-flow frequency or the interpretation of intense precipitation as we apply 21-yr and 150-yr bandwidths for the frequency analyses.

There are several challenges to infer debris-flow magnitudes and frequencies from df turbidites. Subsequent debris flows of larger magnitude can incorporate previous terrestrial debris deposits by sediment entrainment; therefore over time most of the total debris-flow volume ends up in the sediment record of the lake. Df turbidites remain macroscopically undetected if their layer thickness is below the typical lamination thickness of the background sedimentation and in this case would require detailed microfacies investigations. Stacked event layers potentially correspond to multiple debris-flow surges of a long-lasting high-discharge event or to coeval debris-flow activity at different fans during the same event. Thus, identification of the clay cap, representing the post-event deposition of a suspension cloud, is crucial to disentangle stacked event layers. Erosion of underlying sediment is generally negligible, as most df turbidites have a coarse silt to fine sand maximum grain size and the coring sites are located far from the slope break where the erosion potential is highest (Fig. S1). The investigated transect provides first-order insights to spatially investigate underflow deposition of the basin and debris-flow activity of the proximal juvenile fan, but several more cores in transects to other fans would be required for a holistic view on the deposition pattern of each df turbidite.

iii) The presented age–depth model results in a 95 % uncertainty range of a few years and decades at the near-surface sediments and increases to a few centuries deeper down (Table S2). Therefore, there is a strong need to also impart this variable age uncertainty to further analyses such as the cumulative thickness or frequency of df turbidites over time. Based on these relatively larger age errors compared to a varve-based age-depth model, we refrain from detailed comparisons of the Plansee record with other flood deposit archives for discussing potential climatic drivers on debris-flow activity (see Sect. 5.4). This would require microscopic event identification, further [14]C dating and detailed investigations (and counting) of potential annual mixed, organic-clastic varves.

## 5.2 Terrestrial and subaqueous depositional patterns

Our amphibious geomorphic investigation provides evidence of debris-flow activity on two fan deltas in different evolutionary stages. Between May and August 2019, the debris-flow channel on the juvenile fan delta was backfilled and overtopped, followed by avulsion and the formation of a new channel (cf. Haas et al., 2016). The ~8 % sediment delivery ratio indicates high connectivity between the catchment and the subaquatic fan delta. An erosion depth of up to 2.6 m suggests that channel scouring supplies most of the sediment volume entering the lake. Due to the limited observation period of three months, the TLS data is not representative when it comes to longer time periods. Nevertheless, we can deduce that deposits currently located in the depositional area of the juvenile fan delta are rapidly subjected to consecutive transport to the lake. Since the juvenile fan contributes more to the subaquatic sediment deposition compared to the mature fan delta, this fan was chosen as a starting point for the core transect.

The discovery of lobe-shaped deposits with high backscatter signals on the subaquatic delta continuation of the juvenile fan provides evidence for recent debris-flow activity and active fan progradation on the juvenile fan. In addition, subaquatic landslide deposits reveal a second process contributing to event deposition in Plansee.

The mature fan delta mainly experienced terrestrial deposition during the investigated period. The average 10° slope is below the deposition angle of debris (<15°, Grelle et al., 2019), which necessarily forces smaller debris flows to deposit a major share of their initial volume. Comparison of the sediment delivery ratio of the two fan deltas demonstrates that there is a certain proportionality of on- and offshore sedimentation respective to the age and steepness of the fans. Low-magnitude debris flows terminating in a zone of high frictional resistance contribute to the lack of sediment transport into the lake. However, the massive subaquatic volume of the fan delta and debris-flow deposits spilling over the currently active channel levee show that over time, most of the debris-flow material accumulates in the lake basin, and the onshore fan thickness represents only part of the total transported volume. Overprinted lobes of few large debris flows on the active subaquatic fan area suggest that terrestrial deposition dominates recently. From the onshore morphology changes and the bathymetric investigation, we

conclude that in recent times, there is a higher magnitude threshold for debris flows protruding into the lake on the mature fan delta.

## 5.3 Deposition patterns of df turbidites

The majority of the investigated df turbidites have ponding sediment bodies indicating overall underflow deposition (e.g. Gilli et al., 2013) induced by debris-flow activity of the adjacent fans. Besides this general deposition trend, several df turbidites show major or even exclusive deposition either in the distal or in the proximal cores, which demand different interpretations. From the limited perspective of this transect of four cores, we interpret that df turbidites with major deposition in the distal (basin) cores (grey arrows in Fig. 7b) are potentially related to high-energy flows, which bypassed the proximal sites. Alternatively, debris-flow activity occurred on another fan delta and our transect cannot resolve its real source. Exclusive deposition in the two distal basin cores only occurs in a sequence from 1966 to 1976 CE ("D" bracket in Fig. 7b), during which extensive street constructions were carried out at the northern side of the basin. Therefore, we interpret this phase of exclusive distal basin deposition to be related to low-energy, human-induced detrital input from the northern shoreline and thus, df turbidites 11–16 likely reflect anthropogenic impact and not debris flow-induced turbidites. In contrast, 14 df turbidites with a major thickness in the fan-proximal cores (black arrows in Fig. 7b) form wedge-shaped sediment bodies near the slope break and can be explained by low-energy debris-flow activity on the juvenile fan, which may be caused by shifts in the delta morphology (e.g. formation of multiple branching channels) or changes in the connectivity between catchment and depocentre. The linkage of these sediment deposits to the investigated juvenile fan delta reveals that its particular catchment has shown episodic debris-flow activity over at least 4,000 years. Moreover, the sedimentary record holds a short period of exclusively wedge-shaped df turbidite sedimentation ("P" bracket in Fig. 7b), which occurred immediately after the ~2120 BCE earthquake expressed as multiple subaqueous mass-wasting deposits overlain by the eq3 turbidite (Fig. 7a; Oswald et al., 2021). This earthquake was interpreted to trigger the large-scale catastrophic rockslides at Eibsee and Fernpass, both within 15 km distance, and thus can also have triggered small-scale mass movements in the catchment of Plansee, causing an enhanced availability of loose sediment. Such additional sediment availability in a lake catchment can lead to enhanced detrital input in the lake in the aftermath of a strong earthquake ('postseismic landscape response' of Howarth et al., 2016). Accordingly, we interpret the sequence of exclusive proximal event deposition in the aftermath of the ~2120 BCE earthquake to represent highly sediment-concentrated but low energy flows from the adjacent juvenile fan caused by seismically induced enhanced sediment availability in the catchment. Following our observations, it took about 80 years for the landscape to recover and return to equilibrium state after the ~2120 BCE earthquake.

## 5.4 Driving forces of debris-flow activity

The Plansee area offers (i) large sediment availability, (ii) juvenile highly connective fan morphology, (iii) small changes in vegetation cover, and (iv) little human influence in the debris-flow release zones compared to other catchment areas in the European Alps. Thus, the temporal fluctuations are presumably mostly controlled by (v) a few earthquakes and (vi) climate forcing.

i)   The intensely jointed dolomites (Hauptdolomit) surrounding Plansee form a quasi-constant and practically infinite supply of loose debris. In this transport and not weathering-limited setting, the slopes are highly sensitive to short, intense precipitation triggering the release of debris flows. During phases of postseismic landscape response, the precipitation threshold to initiate sediment transport could be lowered (see Sect. 5.3).

ii)  Debris-flow deposition depends on the fan morphology. The prevalent fan type at Plansee displays a highly connective morphology, where debris flows bypass the steep fan delta and deposit offshore. Since most fans are still in the juvenile stage of development, subaquatic deposition of most events is ensured and changes in fan morphology can likely be ruled out as a major controlling factor for changing debris-flow activity over the last 4,000 years.

iii) The percentage of vegetation cover on debris-flow fans is mainly regulated by the debris-flow activity (Dietrich and Krautblatter, 2017). Human-induced vegetation changes are documented since about 1,000 BCE from a pollen record from peat bog remnants at Heiterwanger See a few kilometres west of Plansee (Kral, 1989). A period of enhanced forest clearance in the area happened during medieval times according to this record. The herein presented sedimentary record of Plansee shows no signs of a drastic increase in debris-flow frequency throughout this period of forest use. During late medieval times, the climatic deterioration of the Little Ice Age, war, and epidemics led to a decrease in population and forest clearance, which can be observed in pollen diagrams from the nearby Ostallgäu (Stojakowits and Friedmann, 2013). Since then, no further increase of forest clearance or wildfires has been reported in the area. Therefore, we infer that there were no significant changes of vegetation before or during the period of increasing debris-flow frequency in the last century (df phase 4).

iv) Human interference on Plansee is minor compared to other Alpine lake environments and mainly affects the shoreline and the subaqueous realm. Artificial lake level changes since 1902, wave action from the operation of a cruise ship since 1927, and construction works on the street nearby the northern shoreline may have had an impact mainly on coastal erosion. While shoreline processes contribute to the background sedimentation, it is possible that they also create subaquatic detrital layers which are difficult to distinguish from other event deposits. Therefore, some of the df turbidites in the first half of the 20$^{th}$ century might actually reflect human-induced mass wasting (see also vi). Debris flows can entrain sediment from the coastal zone, which increases the sediment volume reaching the lake basin. The construction of ripraps and retention basins since the mid-20$^{th}$ century possibly led to a slightly increased threshold for sediment delivery into the lake by debris flows, causing the slightly decreased frequency of df turbidites in phase 4.2. However, human interference plays only a subordinate role in altering the process and occurrence rate of debris flows since their zone of release is located well above the human influenced areas i.e. shoreline and subaqueous realm.

v) Strong earthquake shaking causing mass movements can fundamentally change the sediment availability in the catchment and is interpreted as the causal factor for increased debris-flow activity in phase 1 (Sect. 5.3). Such a postseismic increase in debris-flow activity could also be interpreted for eq1 at 1930 CE but in much lower magnitude (~1.3-fold frequency increase) and extent (5-10 years) than for eq3 at ~2120 BCE (80 years). However, conclusive interpretation of postseismic landscape response following eq1 is not possible, due to stacked effects of the contemporary human influence. In contrast, after eq2 at 1050 BCE no significant increase in debris-flow activity can be observed. Eq2 was inferred to be located more to the south and had likely less local intensity at Plansee (Oswald et al., 2021). Thus, we conclude that only the strongest shaking events well above local intensity of VI generate a distinct postseismic landscape response in a lacustrine record. The M 5.3 Namlos 1930 CE earthquake with local intensity of VI½ to VII at Plansee (Oswald et al., 2021) possibly had a minor influence on the sediment availability in the catchment. Given that debris-flow activity increases already at ~1920 CE, we infer that the postseismic landscape response after the 1930 CE earthquake is not the dominant factor influencing the debris-flow activity in the following decades. This is in accordance to other lacustrine studies worldwide, where significantly enhanced fluvial sediment transport to lakes has only been observed for seismic intensities of ~VIII-IX (e.g. Howarth et al., 2016; Moernaut et al., 2014).

vi) Changes in precipitation patterns through climate forcing are the main factor controlling debris-flow activity (Jomelli et al., 2019) and intense rainfall is the most important trigger mechanism for their release. Since the Plansee catchments offer large material supply, the system is highly sensitive to changes in precipitation. Variations in the precipitation pattern are directly reflected in the well preserved and highly resolved sediment archive of the last 4,000 years. Potential temporal overlap of two df turbidites with the two heaviest rainstorms of the century in 1999 and 2005 CE (Fig. 8b) let us infer that regional (advective) rainstorms lasting over several days are a trigger mechanism

of debris flows also in the Plansee region. This is also supported by coincidence of several outstanding thick df turbidites of phase 2 e.g. in the first and eighth century BCE with a period of enhanced flood activity that has also been documented in the record of Ammersee (Czymzik et al., 2013) and Mondsee (Swierczynski et al., 2013). However, for the last century in phase 4, not all rainstorms triggered a macroscopic df turbidite and not each df turbidite has a corresponding rainstorm event (Fig. 8b). This is expected since local storm cells have diameters of a km or less. Moreover, the precipitation sums are recorded on a daily base, where short but extreme convective precipitation is not distinguishable. Potential mismatches might also be due to the age error of df turbidites, with a 95 % probability range from 5 years for the youngest event to 60 years at the beginning of df phase 4. In any case, a relationship between short extreme convective precipitation and debris-flow activity in the Alps is commonly observed in recent times (e.g. Schneuwly-Bollschweiler and Stoffel, 2012).

The region has experienced a 2-fold increase in convective precipitation frequency between 1920 and 2010 CE with an average increase of 10 % per decade (Hydrographischer Dienst Tirol, 2020). The 40 km distant Hohenpeissenberg Meteorological Observatory recorded a 2-fold increase in days with precipitation $\geq 30$ mm from 1879 to 2000 CE (Fricke and Kronier, 2002). Before comparing these rainfall records with our debris-flow record, human influences in the 20th century need to be considered. The presented debris-flow record is likely overestimated in phase 4.1 (~1920 to ~1980 CE) due to artificial lake level changes, coastal erosion, and road constructions, but the record is likely underestimated in phase 4.2 (since ~1980 CE) due to preventive constructions in the northern lake part (see Sect. iv above). Without the possibility of quantifying these human influences, we infer a mean frequency of phase 4.1 and 4.2 to be a best estimate, showing a ~7-fold increase in phase 4.2 compared to phase 3 (~1520 to ~1920 CE) coincident with the instrumentally documented 2-fold increase in rainstorm activity ($\geq 35$ mm d$^{-1}$) from 1920 to 2010 (Dietrich and Krautblatter, 2017). We provide sedimentological evidence for the increased debris-flow activity in the 20th century, as previously observed on differential LiDAR data from several fans at the northern shore of the lake (Dietrich and Krautblatter, 2017). An increasing rainstorm frequency since the 20th century is also observed in several lacustrine flood records in the Alps (e.g. Glur et al., 2013; Swierczynski et al., 2013) and via historically documented river floods in central Europe (Blöschl et al., 2020), pointing to regional changes in the atmospheric circulation patterns. However, the absence in the Plansee record of other historic periods with enhanced rainstorm activity documented in these other records let us infer that debris-flow activity in small catchments is strongly controlled by local high-intensity convective precipitation events. A temperature rise associated with ongoing global warming can cause increased convective rainfall (Diffenbaugh et al., 2013) and therefore likely lead to increased rainstorm-triggered debris-flow activity.

## 6    Conclusion

The sedimentary infill of Plansee holds a well-preserved Alpine archive of Holocene debris-flow activity in a system of high permanent debris production on juvenile, highly connected fan morphologies. Here, we present a 4,000 year continuous record of debris-flow dynamics and an unprecedented amphibious characterization of debris-flow fans. Actualistic debris-flow processes and their corresponding deposits are characterized by their geomorphic landforms in both the terrestrial and subaqueous realms. In a transect of four sediment cores from an active juvenile fan towards the distal basin, we distinguish debris flows and earthquakes related with different turbidites based on their geomorphological, sedimentological and geochemical characteristics. Debris flows form lobe-shaped deposits with high back-scatter signals on the subaquatic prolongation of a subaerial active channel on the fan delta. Df turbidites show a graded grain-size trend, have a less steep D50/D90 evolution and contain more TOC compared to the homogeneous eq turbidites. Df turbidite thickness distributions in the core transect hints at underflow deposition and provides insights into source area and flow energy.

Frequency analyses of df turbidites show that relatively low and constant debris-flow activity over the last 4,000 years is interrupted by i) an increased debris-flow frequency in the aftermath of a local severe earthquake at ~2120 BCE and ii) a fast and high frequency increase since the 20th century, which has not been experienced in the previous record. Numerous empirical studies and global climate models attribute an enhanced hydrological cycle and increase in frequency and/or magnitude of heavy precipitation to climate forcing. Therefore, the temporal coincidence of increasing debris-flow frequency at Plansee with enhanced rainstorm activity in the 20th century provides further evidence for the direct link between climate change and debris-flow activity.

**Data availability**

The TLS data, bathymetric data and grain-size data from this study are available upon request.

**Author contribution**

MK, MS, JM, CM and PO designed the study. PO, CK, MK and CM conducted the sediment core sampling. CK acquired and processed the LiDAR data and interpreted it with help of MK. CK and CM conducted carbon geochemistry analyses. SF and PO acquired the bathymetric dataset. SF processed the bathymetry data and PO geomorphologically interpreted the bathymetry. PO carried out sediment core analyses with help of CK. PO and CK created the figures and wrote the manuscript with input of all co-authors.

**Competing interests**

The authors declare that they have no conflict of interest.

**Acknowledgements**

We thank the team of Freiwillige Feuerwehr Reutte and Wasserrettung Reutte for on-spot support and fieldwork assistance. Gabriel Altebäumer is thanked for his help and support with sediment core sampling and LiDAR measurements. We thank Richard Lipp for providing important historical information. PO, JM, and MS received funding by the Tyrolean Science Fund (project number UNI-0404-2151) the Austrian Science Fund (FWF): P30285-N34, European Regional Development Fund Interreg 312 V-A (project ITAT-301 6) and Austrian Academy of Sciences ÖAW (ESS-IGCP-project S4LIDE-Austria). PO received funding for the bathymetric acquisition through a research grant from the doctoral programme Natural hazards in mountain regions at University Innsbruck. Land Tirol – data.tirol.gv.at is thanked for providing the DEM data. We thank the editor and acknowledge critical comments of three reviewers on a previous version of this manuscript.

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
