# Peer review of "A 4,000 year debris-flow record based on amphibious investigations of fan delta activity in Plansee (Austria, Eastern Alps)"

_Earth Surface Dynamics, 2021_

## Referee Report (RR1)

Dear Editors,
Dear Authors,

Thank you for letting me read and revise this interesting manuscript.

This manuscript presents an extraordinary dataset:
Very interesting new and detailed data on the topography of the alluvial fans and the surface change of these fans over a three month period in combination with bathymetric and backscatter data of the Plansee lake floor. These data quantify the current state of these surfaces and a short term change of the terrestrial part. Secondly, the data comprise sediment cores in which the turbidity-current event stratigraphy has been analyzed over longer time scales (4000 yrs). The paper showcases exciting data, is well written and mostly well illustrated.

However, I do have a major concern regarding the logic behind this paper, that assumes that every turbidite deposited – besides the ones that are attributed to earthquakes from another publication - corresponds 1:1 to a single debris flow event onshore and no other triggers or turbidity currents that occur without a clear trigger are considered. This excludes a bunch of new and often surprising insights from recent turbidity-current monitoring experiments that show that turbidity currents can be triggered even by dilute surface plumes and often occur without a clear trigger. I do agree that this particular system might be dominated by terrestrial debris flows but this does not allow to connect each non-Earthquake triggered turbidite to a single terrestrial debris flow within the past 4000 years. I explain my argument in detail citing the relevant literature below.

However, I still think that there is much value in quantifying the recurrence rate and thickness of such events in the record. Recent literature has shown that these can contain important information on sediment flux and sediment transport processes (citations in main comment below). I suggest to represent this lacustrine turbidite record as a proxy for sediment delivery by sudden events into the basin, rather as a 1:1 terrestrial debris flow record derived from lake turbidites. That includes rephrasing the title of the paper. This will prevent the overinterpretation of the lacustrine turbidite record to a single terrestrial sediment-transport process but still potentially demonstrate the interesting increase in turbidite frequencies and thicknesses in the 20$^{th}$ century.

I also have additional comments on
  1. the robustness of the distinction of 'df turbidites' and 'eq turbidites' based on geochemical and grain-size profiles,
  2. age & turbidite thickness & frequency uncertainties,
  3. Sadler effect-related issues regarding the increase of turbidite frequency and thickness, and
  4. the value of the sediment delivery ratio in this context,
which I also list below.

I recommend major revisions of the manuscript but, at the same time, want to stress the high quality of the data sets and the efforts made for analyses. Once the authors put potential causes of turbidite deposition into a more realistic perspective, this manuscript will make a relevant contribution to the field of geomorphology an sedimentology.

I hope the authors find these comments useful.

Best Wishes,
Anne Bernhardt

**Main comment**

The paper distinguishes between earthquake-induced and debris-flow related turbidites.
The evolvement of terrestrial and subaqueous debris flows into turbidity currents has been well established in experimental data and in field and subsurface studies (Talling, 2013). However, flooding events as triggers of turbidity currents are excluded as no permanent river flows into the lake. However, there are ephemeral streams discharging into the lake that could shed fluvial floods into the basin (e.g., Katz et al., 2015). I do not see a way how one would distinguish between these types of triggers based on the current data set.

The association of debris flows and turbidity currents has been shown in marine strata where marker beds and continuous outcrops have allowed bed-by-bed tracing (Amy and Talling, 2006). However, a clear association of debris flow deposits to equivalent turbidity currents has not been shown by the authors of this paper.

Moreover, turbidity currents do not necessarily need a specific trigger, piling up of loose sediment on the lake-margin slope and excess pore pressure will eventually result in failure and a turbidity current can evolve. Frequent turbidite activity with no major triggers has been shown by direct monitoring in the recent years (e.g., Clare et al., 2016; Paull et al., 2018). Furthermore, Hizzett et al. (2017) have shown that turbidity currents are most often triggered by settling plumes, rather than landslides or flooding events. I urge the authors to familiarize themselves with these recent research on triggering of turbidity currents, which has led to very surprising results and shifting paradigms over the recent years. I do agree that these experiments were carried out in different settings that the Plansee area, but they do show the complexity of turbidity current triggering.

It seems like the authors are 'automatically' categorizing turbidity currents as debris-flow induced when they were not classified as earthquake-triggered in a previous publication by Oswald et al. (2021). In line 447 and onwards, they state: "The potential of misinterpreting df turbidites as river flood-induced turbidites, which could have similar characteristics (Gilli et al., 2013; Wilhelm et al., 2013), is very low at this subbasin of Plansee. This is because possible hyperpycnal flows related to the main inflowing rivers are trapped either in Heiterwangersee or in the easternmost subbasin in Plansee and do not reach the studied main basin". I do not think that this holds true, turbidity currents in the marine realm can cross obstacles of several 100s of meters height and have run out distances of > 1000 km on very low slopes (Talling et al., 2007; Völker et al., 2008). They can sure pass from the easternmost subbasin into the main basin, a distance of only < 4km (but not from the Heiterwangersee to the Plansee basin, I agree). Surely, lacustrine turbidity currents are probably smaller than big marine events, but there is no basis to exclude that these run across the lake floor. Plus, turbidity currents – again - can occur without specific triggers.
I do agree that based on the abundance of debris flow lobes on the modern subaqueous fan delta, as nicely illustrated in the first part of the manuscript, debris flows are a very important sediment transport process in this area. However, given the complexity of subaqueous sediment failure in natural systems, one cannot assume in turn that every single turbidite (not assigned to an earthquake) has been triggered by a debris flow for the past 4000 years. As mentioned before, data from recent experiments that monitored turbidity currents directly also on deltas – do not show much evidence for this at all.

In order to go forward with this interesting record, I would distinguish between earthquake-induced and other turbidites. There is still lots of value in quantifying turbidite frequency and thickness as a general proxy for sediment input related to episodic sediment density flows (e.g., Covault et al., 2010; Bernhardt et al., 2017). Supposedly, many of these turbidites are related to debris flow events, but by no means can this record be used as a direct record for (onshore) debris flows only.
The title would need to be adapted.

The discussion section 5.4 should be reframed (and shortened) to 'Driving forces/ controls on turbidity-currents activity' and carefully avoiding overinterpretation of the record.

**The robustness of the distinction of 'df turbidites' and 'eq turbidites' based on geochemical and grain-size profiles**

Line 715 of the Conclusion states:

"Df turbidites show a graded grainsize trend, have a less steep D50/D90 ratio and contain more TOC compared to the homogeneous eq turbidites. „

In my mind this statement is not fully supported by the data shown in Figure 6.

The two distinctly looking grain size profiles of Fig 6a obviously do not prove that there is a statistical difference between the two types (if there are only 2 types of tubiidites, see my doubts above…) in terms of the grain-size grading of the deposit. The authors need to show that there is a robust difference in the grading of the two deposit-groups based on the analyses of many deposits.

In my view Fig. 6b shows no significant difference between the different turbidites types in TOC-d13C space. If the authors want to state a difference here, they need to show by e.g., a cluster analysis, that these clusters are actually statistically distinct and the two groups cannot be drawn from a single distribution. How does the sample size (many df , fewer eq and background data points) affect the analyses? The same applies to the data shown in figure 6c: Is the difference in D50/D90 space significant? How would acquiring more eq data points affect the result?

**Age & turbidite thickness & frequency uncertainties**

Unclear how the [137]Cs & [210]Pb data informed the BACON model in Fig S2. Did it?

The authors address the uncertainties in the age models by showing estimates for every single BACON simulation, a valid approach in my mind. However, this approach is only shown for the 150 yr bandwith, not for the 21 yrs. Hence, the uncertainties on this model cannot be assessed. Please add.

Can you comment on the uncertainties on (often very thin) turbidite thickness measurements due to the often diffuse upper boundary to the background sedimentation?

Moreover Table 1 shows a bunch of parameters for each 'phase' with no uncertainty estimates. These need to be added to make this comparison between 'phases' meaningful.

Difference in Phase 3 and 4 Fig 8a: Is the increase in cumulative thickness between phase 3 and 4 actually significant when the uncertainties are accounted for? Cannot tell from this plot.
Line 695 and ongoing:

„we infer a mean frequency of phase 4.1 and 4.2 to be a best estimate, showing a ~7-fold increase in phase 4.2 compared to phase 3 (~1520 to ~1920 CE) coincident with the instrumentally documented increase in rainstorm activity"

Interesting, but I cannot follow this argument as the rainfall data are not shown in your figures as only the precipitation record of the past 118 years is shown in figure 8b. This is an important observation, to convince your reader you need to show the data & the robustness of the correlation.

**Sadler effect-related issues regarding the increase of turbidite frequency and thickness**

Turbidite thickness and deposition rate is generally increasing towards younger deposits. Can these changes be simply attributed to the statistical bias describes as the Sadler effect as older strata is more prone to hiatuses and sediment removal (Sadler, 1981; Sadler and Jerolmack, 2014).

**The value of the sediment delivery ratio in this context**

Line 188, Terrestrial LIDAR data Equation 1

I do not understand how you determine  the numerator: sediment flux into the lake, by 3 LiDAR surveys which are 3 month apart.  What about the sediment flux from farther up the catchment that bypassed the survey area and ended up in the lake? Please clarify.

Line 395 states: During the 3 months, the sediment only accumulated on the terrestrial fan delta.

How do you know? Differential bathymetry on the lake bottom? Sediment traps employed? How do you now no sediment bypassed into the lake?
I guess your argument is the sediment delivery ratio. But (if I understand this correctly) that ratio is only based on data of the fan itself. However, material mobilized further up in the catchment could have simply bypassed the fan (eroded some, deposited some on the alluvial fan) and deposited in the lake. I do not think this ratio is useful to determine how much sediment made it into the lake during the 3-month period, if not the entire catchment draining into the lake is imaged & considered.

And one more:
Line 577 I am not sure you can use the sediment delivery ratio – the way you calculated this – as a measure for system connectivity, as there might be much more sediment involved from further up in the catchment making it into the that is not accounted for by your difference maps of the fan. I don't think this does much to your story. Would just leave this out and shorten the manuscript somewhat.

**Landscape recovery from earthquakes**
This is a very interesting aspect that may deserve more attention in some other manuscript. However , if I read figure 7a correctly – events during the "P-period' can only be recognized in the proximal cores Pan 19-03 and Plan19-04 because the two distal cores (Plan18-10 and Plan 19-02) do not penetrate deep enough to even capture these events. So how do you know these events are not recorded in these locations but were simply note penetrated by the two distal cores?

**Minor comments:**

It sure is a challenge to derive useful information of datasets covering completely different time scales. The 'amphibious' approach is much praised in the intro, however, what are exactly the new insights that we gain form this approach? This falls somewhat short.
In my view you can show that the currently active parts of the alluvial fan-delta system, that   active delta front is dominated by debris flow lobes and erosion and deposition happens on the alluvial fan.

In line 98, the authors state that there is no permanent river inflow to the main basin of the Plansee. However, Line 112 mentions two permanent inflows /draining catchments) into the lake. Please clarify the discrepancy. By main basin you mean your specific study area? And these basins are well separated by a high morphologic barrier in between that turbidity currents cannot cross? Does not look like it in your figure 1…

Fig. 3 and line 590
What do you mean by overspilling debris flows. Spilling over what? Beyond the break of slope onto the lake floor?

Line 119 states that
„While the human interference may influence the continuous background sedimentation in the lake quite a bit, the episodic debris flows eroding materials from 10,000 m2 large catchments way above the lake will not be relevantly influenced." How are you so sure? Is there no human activity at all in the catchments above the lake? Deforestation during historic times/ reforestation after etc?

Line 300 and ongoing.
It seems like the authors assume that debris flows are the only sedimentary process happening on an alluvial fan. What's your evidence for that? Are there no fluvial or sheet flow processes happening? 100% debris flow? Are these deeply enough incised (4000 yrs of stratigraphy) and these outcrops show only debris flow deposits and no fluvial processes?

Line 410
The Lightness L* factor of the sediment is first mentioned here but should be explained in the Methods section already

Avoid unnecessary repetition of words throughout the manuscript, e.g.,
Line 42 two times "worldwide"
Line 16 and 18: "amphibious" (is this a good word for what you actually mean – combined terrestrial and lacustrine data acquisition)

**Figure 2d and others.**
To ensure inclusiveness, make sure your figures are legible to people with certain color-blindness ( 8% of all men and 0.5% of all women are colorblind). I am not colorblind (I think ;)  but the differences in color are hard for me to see. I used this website to check your Fig 2d
https://www.color-blindness.com/coblis-color-blindness-simulator/
and apparently the figure looks like this for people with Red-Blind/Protanopia & your differently mapped areas are undisguisable.
Enhance the contrast between the colors and this will work better.

[Figure]

References:

Amy, L.A., and Talling, P.J., 2006, Anatomy of turbidites and linked debrites based on long distance (120 x 30 km) bed correlation, Marnoso Arenacea Formation, Northern Apennines, Italy: Sedimentology, v. 53, no. 1, p. 161–212, doi: 10.1111/j.1365-3091.2005.00756.x.

Bernhardt, A., Schwanghart, W., Hebbeln, D., Stuut, J.-B.W., and Strecker, M.R., 2017, Immediate propagation of deglacial environmental change to deep-marine turbidite systems along the Chile convergent margin: Earth and Planetary Science Letters, v. 473, doi: 10.1016/j.epsl.2017.05.017.

Clare, M.A., Hughes Clarke, J.E., Talling, P.J., Cartigny, M.J.B., and Pratomo, D.G., 2016, Preconditioning and triggering of offshore slope failures and turbidity currents revealed by most detailed monitoring yet at a fjord-head delta: Earth and Planetary Science Letters, v. 450, p. 208–220, doi: 10.1016/j.epsl.2016.06.021.

Covault, J.A., Romans, B.W., Fildani, A., McGann, M., and Graham, S.A., 2010, Rapid Climatic Signal Propagation from Source to Sink in a Southern California Sediment-Routing System: The Journal of Geology, v. 118, no. 3, p. 247–259, doi: 10.1086/651539.

Hizzett, J.L., Hughes Clarke, J.E., Sumner, E.J., Cartigny, M.J.B., Talling, P.J., and Clare, M.A., 2017, Which triggers produce the most erosive, frequent and longest runout turbidity currents on deltas? Geophysical Research Letters, p. 1–9, doi: 10.1002/2017GL075751.

Paull, C.K., Talling, P.J., Maier, K.L., Parsons, D., Xu, J., Caress, D.W., Gwiazda, R., Lundsten, E.M., Anderson, K., Barry, J.P., Chaffey, M., O'Reilly, T., Rosenberger, K.J., Gales, J.A., et al., 2018, Powerful turbidity currents driven by dense basal layers: Nature Communications, v. 9, no. 1, p. 1–9, doi: 10.1038/s41467-018-06254-6.

Sadler, P., 1981, Sediment Accumulation Rates and the Completeness of Stratigraphic Sections: The Journal of Geology, v. 89, no. 5, p. 569–584.

Sadler, P.M., and Jerolmack, D.J., 2014, Scaling laws for aggradation, denudation and progradation rates: the case for time-scale invariance at sediment sources and sinks: Geological Society, London, Special Publications, v. 404, p. SP404.7, doi: 10.1144/SP404.7.

Talling, P.J., 2013, Hybrid submarine flows comprising turbidity current and cohesive debris flow: Deposits, theoretical and experimental analyses, and generalized models: Geosphere, , no. 3, p. 1–29, doi: 10.1130/GES00793.1.

Talling, P.J., Wynn, R.B., Masson, D.G., Frenz, M., Cronin, B.T., Schiebel, R., Akhmetzhanov,  a M., Dallmeier-Tiessen, S., Benetti, S., Weaver, P.P.E., Georgiopoulou,  a, Zühlsdorff, C., and Amy, L. a, 2007, Onset of submarine debris flow deposition far from original giant landslide.: Nature, v. 450, no. 7169, p. 541–4, doi: 10.1038/nature06313.

Völker, D., Reichel, T., Wiedicke, M., and Heubeck, C., 2008, Turbidites deposited on Southern Central Chilean seamounts: Evidence for energetic turbidity currents: Marine Geology, v. 251, no. 1–2, p. 15–31, doi: 10.1016/j.margeo.2008.01.008.

---

## Author Response (AR3)

Dear editors and referees,

we want to thank the editor and both referees for their insightful and detailed reviews. We revised the manuscript addressing their comments and added a discussion of the caveat brought forward. A point-by-point response to the comments is provided below, structured as follows: (1) referee comment, (2) author's response and (3) changes in manuscript.

Many thanks and best regards,
Carolin Kiefer, on behalf of all authors

**(1) Comment by Associate Editor:**

Dear authors,

first, let me apologise for the long time it took to deliver these reviews and decision. It was not easy to find suitable reviewers, and there were delays in the correspondence, likely due to the holiday season.

In any case, I received two reviews. The first of these was delivered by one of the original reviewers, who is very happy with your revisions and has only minor comments. The second was by a new reviewer, who raises an additional caveat, namely that the turbidity events you identify in the lake records not necessarily correspond to debris flow events onshore in a one-to-one manner. I think this is a valid point. However, it seems to me that this notion will not majorly change your results, especially given that you use a smoothing filter with a bandwidth of 150 years to assess the temporal evolution of event frequency. Still, I think the caveat needs to be explicitly pointed out and discussed in the paper.

Given this assessment, I recommend minor revisions. Please supply a detailed rebuttal with your revised manuscript. Depending on the quality of the response and revisions, I may decide to send the manuscript out for a re-review by Reviewer #2 upon resubmission.

Looking forward to reading your revised manuscript!
All the best wishes, Jens Turowski

(2) We would like to sincerely thank the editor for the open and prompt communication, for his effort in finding knowledgeable reviewers and for the opportunity to further improve our manuscript. We appreciate his comment on the conclusive assignment of triggering processes grounding on the main comment of the reviewer #3 which we now fully address in the revised manuscript. In particular, we solved this caveat and improved the manuscript by i) further explanation of the peculiar setting at the main basin of Plansee, ii) presenting the two-fold categorization of turbidites as first-order interpretation and initial hypothesis, which we want to test in this study, and iii) interpretation and justification of the event categorization in the discussion section.

Besides addressing this main comment, we also fully address all minor comments raised by the reviewers. We now hope that this revised manuscript meets the high-quality standards for publishing our research in Earth Surface Dynamics.

**Report #1**
**Submitted on 23 Aug 2021**
**Anonymous Referee #1**

(1) The revised version of the manuscript esurf-2021-23 by Kiefer et al. presents a thorough revision of the original manuscript. The authors invested a great deal of effort to address all my previous questions and answered almost all of them to my full satisfaction. I still have several questions and remarks concerning the event frequency analysis. I believe the manuscript will be suitable for publication in Earth Surface Dynamics after these have been addressed.

(2) We thank the reviewer for this positive feedback on our first revision and also for the additional comments especially on the presentation of the quantiles, which we implemented in the revised manuscript. Besides this, we fully address all minor comments by the reviewer in the revised manuscript (see replies below).

(1) Remark:

I am not sure if I understood the argument of the authors against the sampling theorem in this case. I fully agree with the authors, that lacustrine sedimentation can be regarded as continuous for frequencies above the annual cycle. Since the underlying (continuous) age-depth function is estimated with (discrete) radiocarbon samples, the sampling theorem must be applicable in my opinion. Nevertheless, I am very happy with the results from the individual Bacon simulations, which I believe increase confidence in the applied bandwidth of 150 years. I only have a few additional comments/questions:

a) I don´t see any value in presenting the results from the 21-year bandwidth estimation. It is not suited to make comparisons between df occurrences in the last two centuries and the rest of the record, since it underestimates occurrence rates in df phases 3,2, and 1. A more informative approach could involve calculating quantiles (e.g. 0.975, 0.5, and 0.025) from the individual simulations (especially for illustration in Fig. 8). Especially the 0.5 quantile could subsequently be used for inter-record comparisons.

b) Please indicate which age-depth model was used to calculate the optimal bandwidth (Bacon mean scenario?).

c) Please indicate, how the frequency analysis in the 150-year window was carried out at the edges of the record (e.g. by cutting off 75 years at each end or reducing the window size near the edges?).

(2)

ad a) We fully agree with the reviewer that the 21-year running sum is probably oversampled for the df phases 3 and 2 following the sample theorem and thus, both under- and overestimates the real frequency there. However, following this sample theorem and the reviewer's suggestion to make use of a bandwidth selection, the use of the 21-year bandwidth for the last century is justified i.e. by the increased occurrence of df-turbidites and increased amount of age constraints ($^{210}$Pb and $^{137}$Cs dates, coring age and $^{14}$C ages). Thank you for suggesting the calculation of quantiles, which we applied to the 150-yr and 21-yr bandwidths of the df turbidite frequency resulting in a 95% frequency probability. We addressed this comment by revising Figure 8a, adding a supplementary Figure S3 to provide the frequency probabilities for the whole record although they are over- or undersampling the event catalogue for certain periods and revised the text accordingly in Lines 283–285; 288–291; 514–522; Supplement Lines 13–18.

(3)

>Lines 283–285

"We applied this bandwidth on the occurrence rate calculation for each individual simulation of the age-depth model derived from the R-software Bacon v 2.4.3 (Blaauw and Christen, 2011) and calculated the quantiles of the yearly df turbidite occurrence rate (0.025, 0.5 and 0.975) to obtain a 95% frequency probability."

>Lines 288–291

"To account for a higher resolution in frequency changes especially in periods with higher number of events, we also calculated a bandwidth based on the df occurrence of the last two centuries and also applied the resulting 21-yr bandwidth to occurrence rate calculation of the main age and 95% frequency probability of all individual age-depth model simulations."

>Lines 514–522

"On top, the annual occurrence rate of df turbidites is displayed as the annual frequency of the mean age (21-yr and 150-yr bandwidth) and the 95% frequency probability bracketed by the 0.025 and 0.975 quantiles of all 6396 individual age-depth model simulations (150-yr bandwidth: light-brown;

21-yr bandwidth: grey). Note that the 21-yr and 150-yr bandwidth frequencies are only valid for certain periods defined by a bandwidth selection test and are over-/undersampled for other periods. A comparison of the frequency probabilities from the two bandwidths is provided in Fig. S3. The four df phases (phases of similar debris-flow activity derived from a change-point analysis of interevent times) are delimited by vertical dashed lines across these three plots. The grey-shaded inlet on top shows a zoom of the cumulative thickness and 21-yr and 150-yr frequency of debris flows in the period 1700–2018 CE. Note the good fit of the yearly df turbidite frequency from the 0.5 quantile with the mean age from the 21-yr bandwidth in the last century."

>Supplement Lines 13–18

"Fig. S3: Complete documentation of the annual df turbidite frequency showing the 150-yr bandwidth (top) and 21-yr bandwidth (bottom), the mean age derived from the age-depth model and the 95 % range of frequency probability derived from the calculation of quantiles (0.025 and 0.975) on all individual age-depth model simulations. Based on the bandwidth selection test after Sheather & Jones (1991) the 21-yr and 150-yr frequencies are only valid for certain periods and are over- or undersampled for the other periods. Therefore, the time periods assigned as 'invalid' should not be considered and are herein only displayed for a complete data presentation."

(2) ad b) We added the information that the bandwidth selection is based on the mean age-depth model to the Methodology (see Lines 281–283).

(3)

>Lines 281–283

"First, a suitable bandwidth (150 yrs) was selected based on the average df turbidite occurrence over the entire core using the mean age-depth model (Sheather and Jones, 1991)."

(2) ad c) We added a sentence to the Methodology section stating that the window size is continuously reduced at the edges (see Lines 285–286).

(3)

>Lines 285–286

"The actual window size for the running sum is continuously reduced at both edges once the sample is smaller than half of the selected bandwidth."

**Report #2**
**Submitted on 30 Aug 2021**
**Referee #3: Anne Bernhardt, anne.bernhardt@fu-berlin.de**

Dear Editors,
Dear Authors,

Thank you for letting me read and revise this interesting manuscript.

This manuscript presents an extraordinary dataset:
Very interesting new and detailed data on the topography of the alluvial fans and the surface change of these fans over a three month period in combination with bathymetric and backscatter data of the Plansee lake floor. These data quantify the current state of these surfaces and a short term change of the terrestrial part. Secondly, the data comprise sediment cores in which the turbidity-current event stratigraphy has been analyzed over longer time scales (4000 yrs). The paper showcases exciting data, is well written and mostly well illustrated.

However, I do have a major concern regarding the logic behind this paper, that assumes that every turbidite deposited – besides the ones that are attributed to earthquakes from another publication - corresponds 1:1 to a single debris flow event onshore and no other triggers or turbidity currents that occur without a clear trigger are considered. This excludes a bunch of new and often surprising insights from recent turbidity-current monitoring experiments that show that turbidity currents can be triggered even by dilute surface plumes and often occur without a clear trigger. I do agree that this particular system might be dominated by terrestrial debris flows but this does not allow to connect each non-Earthquake triggered turbidite to a single terrestrial debris flow within the past 4000 years. I explain my argument in detail citing the relevant literature below.

However, I still think that there is much value in quantifying the recurrence rate and thickness of such events in the record. Recent literature has shown that these can contain important information on sediment flux and sediment transport processes (citations in main comment below). I suggest to represent this lacustrine turbidite record as a proxy for sediment delivery by sudden events into the basin, rather as a 1:1 terrestrial debris flow record derived from lake turbidites. That includes rephrasing the title of the paper. This will prevent the overinterpretation of the lacustrine turbidite record to a single terrestrial sediment-transport process but still potentially demonstrate the interesting increase in turbidite frequencies and thicknesses in the 20th century.

I also have additional comments on
1. the robustness of the distinction of 'df turbidites' and 'eq turbidites' based on geochemical and grain-size profiles,
2. age & turbidite thickness & frequency uncertainties,
3. Sadler effect-related issues regarding the increase of turbidite frequency and thickness, and
4. the value of the sediment delivery ratio in this context,
which I also list below.

I recommend major revisions of the manuscript but, at the same time, want to stress the high quality of the data sets and the efforts made for analyses. Once the authors put potential causes of turbidite deposition into a more realistic perspective, this manuscript will make a relevant contribution to the field of geomorphology an sedimentology.

I hope the authors find these comments useful.

Best Wishes,

Anne Bernhardt

(1) Main comment
The paper distinguishes between earthquake-induced and debris-flow related turbidites.
The evolvement of terrestrial and subaqueous debris flows into turbidity currents has been well established in experimental data and in field and subsurface studies (Talling, 2013). However, flooding events as triggers of turbidity currents are excluded as no permanent river flows into the lake. However, there are ephemeral streams discharging into the lake that could shed fluvial floods into the basin (e.g., Katz et al., 2015). I do not see a way how one would distinguish between these types of triggers based on the current data set.
The association of debris flows and turbidity currents has been shown in marine strata where marker beds and continuous outcrops have allowed bed-by-bed tracing (Amy and Talling, 2006). However, a clear association of debris flow deposits to equivalent turbidity currents has not been shown by the authors of this paper.
Moreover, turbidity currents do not necessarily need a specific trigger, piling up of loose sediment on the lake-margin slope and excess pore pressure will eventually result in failure and a turbidity current can evolve. Frequent turbidite activity with no major triggers has been shown by direct monitoring in the recent years (e.g., Clare et al., 2016; Paull et al., 2018). Furthermore, Hizzett et al. (2017) have shown that turbidity currents are most often triggered by settling plumes, rather than landslides or flooding events. I urge the authors to familiarize themselves with these recent research on triggering of turbidity currents, which has led to very surprising results and shifting paradigms over the recent years. I do agree that these experiments were carried out in different settings that the Plansee area, but they do show the complexity of turbidity current triggering.

It seems like the authors are 'automatically' categorizing turbidity currents as debris-flow induced when they were not classified as earthquake-triggered in a previous publication by Oswald et al. (2021). In line 447 and onwards, they state: "The potential of misinterpreting df turbidites as river flood-induced turbidites, which could have similar characteristics (Gilli et al., 2013; Wilhelm et al., 2013), is very low at this subbasin of Plansee. This is because possible hyperpycnal flows related to the main inflowing rivers are trapped either in Heiterwangersee or in the easternmost subbasin in Plansee and do not reach the studied main basin". I do not think that this holds true, turbidity currents in the marine realm can cross obstacles of several 100s of meters height and have run out distances of > 1000 km on very low slopes (Talling et al., 2007; Völker et al., 2008). They can sure pass from the easternmost subbasin into the main basin, a distance of only < 4km (but not from the Heiterwangersee to the Plansee basin, I agree). Surely, lacustrine turbidity currents are probably smaller than big marine events, but there is no basis to exclude that these run across the lake floor. Plus, turbidity currents – again - can occur without specific triggers.
I do agree that based on the abundance of debris flow lobes on the modern subaqueous fan delta, as nicely illustrated in the first part of the manuscript, debris flows are a very important sediment transport process in this area. However, given the complexity of subaqueous sediment failure in natural systems, one cannot assume in turn that every single turbidite (not assigned to an earthquake) has been triggered by a debris flow for the past 4000 years. As mentioned before, data from recent experiments that monitored turbidity currents directly also on deltas – do not show much evidence for this at all.

In order to go forward with this interesting record, I would distinguish between earthquake-induced and other turbidites. There is still lots of value in quantifying turbidite frequency and thickness as a general proxy for sediment input related to episodic sediment density flows (e.g., Covault et al., 2010; Bernhardt et al., 2017). Supposedly, many of these turbidites are related to debris flow events, but by no means can this record be used as a direct record for (onshore) debris flows only.
The title would need to be adapted.

The discussion section 5.4 should be reframed (and shortened) to 'Driving forces/ controls on turbidity-currents activity' and carefully avoiding overinterpretation of the record.

(2)

We want to thank the reviewer for this thorough and extensive review. We highly appreciate the input on process understanding for sediment transport from deltas into subaqueous (mainly marine fjords) depocenters and also providing relevant literature on this topic. We were already fully aware of the innovative turbidity current monitoring techniques developed at fjord head deltas, but are not convinced that these results and new paradigms can be directly applied to our very different lacustrine study case (see reasoning below). We agree with the reviewer that we cannot pinpoint each individual turbidite conclusively to its triggering mechanism, yet given the detailed explanation and reasoning below, we are convinced that the two-fold categorization of turbidite deposits in the main basin of Plansee is actually justified. Thanks to the reviewer's critical comments, we realized that in the initial manuscript this categorization and justification was insufficiently explained and inadequately presented as a hypothesis to be tested with this study. Therefore, also based on the suggestion of the editor, we added a paragraph in the discussion (Lines 575–598), improved the explanation of our study site turbidite processes therein (Lines 113–117) and presentation of the hypothesis we want to test with this study (Lines 265–269).

The two major points raised by the reviewer in her main comment are 1) that turbidity currents do not necessarily need a specific trigger and 2) that the river inflow from the easternmost basin might also be a potential source of turbidity currents for the main basin of Plansee.

ad 1) The reviewer´s suggestion to consider that turbidites observed in the Plansee cores could correspond to processes that can initiate turbidity currents by slope failures resulting from piling up of loose sediment on the lake-margin slope and excess pore pressure, can be neglected for the studied small-scale lacustrine environment because of the following 3 reasonings:

i)      In intramountainous glacigenic lakes, as Plansee is, lake margin-slopes outside the active fan deltas are very steep and typically-sediment starved (e.g. Strasser et al., 2020). Piling up of sediment on subaqueous slopes only occurs on the subaqueous fan deltas which are genetically connected to the subaerial debris-flow dominated alluvial fans (Dietrich & Krautblatter 2017). There, sediment transport from the ephemeral streams to the subaquatic slopes basin mostly occurs by debris flows initiated by high-intensity precipitation, while sediment discharge during regular precipitation and seasonal snowmelt is comparably low (e.g. Irmler et al., 2006).

ii)     For the main basin of Plansee, we interpreted that the subaerial debris flows continue as subaqueous flows to reach the basin (i.e. our df turbidites). We agree with the reviewer that this 1:1 correspondence might not be entirely correct, because not every subaerial debris flow may reach the basin's depocenter (as also shown and discussed in the manuscript in sections 4.3.2 and 5.3 and Figure 7). However, even debris flows that only reach the alluvial fans influence the stability of the active fan delta slope by sediment loading and thus also reduce subaqueous slope instability. This gives us confidence that most (if not all) non-earthquake triggered turbidites in this peculiar setting actually are related to debris flow events by a) either directly linking onshore sediment transport by debris flow into the lake evolving to turbidity currents in the distal basin, or b) subaqueous debris flow fan delta instabilities immediately or shortly after an onshore debris flow event.

iii)    These processes also result in the active fan delta to be predominantly composed of granular deposits (sand and gravels; see also backscattered data of Figures 2 and 3, and the steep slopes of 30 degree, i.e. close to the angle of response for granular material), while the fine-grained

(< fine sand) sediment are transported downslope by repeated debris flows and debris-flow related fan-slope instabilities (see ii above). On such granular, high-permeable fan delta slopes, excess pore pressure cannot build up significantly over longer time periods, therefore overpressure and low effective stress conditions that would initiate failure are unlikely to occur in this setting.

ad 2) First, it is important to note that the sediment processes occurring on the referred-to river deltas in fjords (with high fluvial sediment input) and the referred-to passive and active continental margins (Talling et al 2007, Völker et al 2008) are not entirely applicable to the studied lake environment in a one-to-one manner: In particular, we disagree with the comparison of turbidity currents offshore NW-Africa and offshore Chile (where flows with tremendous flow-heights and sediment volumes overcome 100's of meters obstacle heights after having travelled downslope several km of water depths and distances of 100's of km) with the small scale lacustrine turbidites at Plansee. For turbidity currents initiated by river floods or delta head instabilities at the river inflow to the easternmost Plansee basin (with maximum water depth of 65 m), we consider it as physically impossible that a turbidity current could gain enough energy along the only 65 m vertical descent and 4 km travel to overcome several hills with >20 m vertical relief to eventually reach the main basin of Plansee.

Most importantly, however, even if we were potentially misinterpreting one or the other df turbidite that still might be related to some other unknown trigger, such few wrongly interpreted turbidites have not a major impact on the inferred debris-flow frequency or the interpretation on intense precipitation as we apply 21-yr and 150-yr bandwidths for the frequency analyses.

In conclusion, we are convinced the caveat (the non 1:1 correspondence of df turbidites with onshore debris flows) raised by the reviewer does not jeopardize the main conclusions of our paper. However, we do acknowledge that applying different monitoring techniques as in the studies referred-to by the reviewer, would allow to even better understand sediment transport in such Alpine lake settings. In this way, we are convinced that this reviewer's comment supports the value of our new amphibious, quasi-4D approach on better understanding the actual transportation and sedimentation processes from onshore debris flows to the turbidite in the subaqueous depocenter. Future projects in such debris-flow dominated lacustrine sites are planned to implement a wide suite of monitoring techniques to study the sediment transport processes in great detail.

Strasser, M., Berberich, T., Fabbri, S. C., Hilbe, M., Huang, J.-J. S., Lauterbach, S., Ortler, M., Rechschreiter, H., Brauer, A., Anselmetti, F., and Kowarik, K.: Geomorphology and event-stratigraphy of recent mass-movement processes in Lake Hallstatt (UNESCO World Heritage Cultural Landscape, Austria), Geological Society, London, Special Publications, 500, 405–426, https://doi.org/10.1144/SP500-2019-178, 2020.

Dietrich, A. and Krautblatter, M.: Evidence for enhanced debris-flow activity in the Northern Calcareous Alps since the 1980s (Plansee, Austria), Geomorphology, 287, 144–158, https://doi.org/10.1016/j.geomorph.2016.01.013, 2017.

Irmler, R., Daut, G., and Mäusbacher, R.: A debris flow calendar derived from sediments of lake Lago di Braies (N. Italy), Geomorphology, 77, 69–78, https://doi.org/10.1016/j.geomorph.2006.01.013, 2006.

Talling, P.J., Wynn, R.B., Masson, D.G., Frenz, M., Cronin, B.T., Schiebel, R., Akhmetzhanov, a M., Dallmeier-Tiessen, S., Benetti, S., Weaver, P.P.E., Georgiopoulou, A, Zühlsdorff, C., and Amy, L.A.: Onset of submarine debris flow deposition far from original giant landslide.: Nature, 450, 7169, 541–4, doi: 10.1038/nature06313, 2007.

Völker, D., Reichel, T., Wiedicke, M., and Heubeck, C.: Turbidites deposited on Southern Central Chilean seamounts: Evidence for energetic turbidity currents: Marine Geology, v. 251, no. 1–2, p. 15–31, doi: 10.1016/j.margeo.2008.01.008, 2008.

(3)

>Lines 113–117

"The herein investigated main (westernmost) basin of Plansee is isolated by circa 20 m high morphological barriers from the circa 4 km distant permanent river inflows in the east, which protects this basin from the influence of river flood-induced turbidity currents. Therefore, the main basin forms a rather unique setting to study debris flows and their related subaqueous deposits, as it is expected that detrital sediment is almost exclusively supplied by episodically occurring debris flows."

>Lines 136–139

"Furthermore, the sedimentary sequence in the main basin of Plansee can also archive extreme sediment transport events in high resolution and high continuity, which sets the stage for the herein presented study on the identification of debris flow-related turbidites and their frequency over time."

>Lines 265–269

"All other event deposits were not the focus of detailed analyses of that study. Given the setting of the main basin with no river inflow, but surrounded by numerous debris-flow fans (Sect. 2), it can be hypothesized that these event deposits almost exclusively are debris-flow related. Thus, this study presumes this two-fold classification of lacustrine event deposits and tests this hypothesis by combining the amphibious geomorphological analyses, turbidite thickness distribution and sediment core analyses."

>Lines 575–598

"The discrimination of the underlying process for different types of turbidites can be challenging especially in clastic influenced lake settings, where pure terrestrial input and remobilisation of subaqueous slopes only show subtle sedimentological and geochemical differences (e.g. Vandekerkhove et al., 2020). Yet, for the main basin of Plansee the initial hypothesis of a two-fold event categorization differentiating eq turbidites and df turbidites (outlined in Sect. 3.3.3) is supported and justified by the data obtained from this peculiar setting investigated by our amphibious geomorphological and sedimentological analyses, as follows: The main basin has no permanent river inflow and is surrounded by numerous alluvial fans, which are dominated by debris-flow activity. A misinterpretation of df turbidites with flood-induced turbidites that would link to the permanent river inflows in the east, can be excluded, as river-flood induced turbidites are deposited in the 65 m deep main depocenter of the eastern basin, while the main basin is protected from the river deltas by circa 20 m high morphological barriers. Eq- and df-turbidites have distinct and distinguishable sedimentological characteristics in Plansee especially on the basis of sediment color (L*), grainsize evolution, organic content and their distribution pattern. Other potential mechanisms that could lead to a similar sedimentological signature as df turbidites are subaqueous fan delta failures initiated by excess pore pressure or sediment loading, which cannot fully be excluded but are unlikely at Plansee. The subaquatic slopes of the active fan deltas mainly consist of gravels and sands (see backscatter data of Figs. 2 and 3), the high permeability of which does not allow excess pore pressure to develop. Some subaerial debris flows might not reach the subaqueous depocenter but add external sediment load on the fan delta which can subsequently initiate failure of the active fan delta slope. There might be a time delay between the subaquatic slope failure and the external sediment loading by the subaerial debris flow, or fan delta sediment remobilization may also only be initiated by a subsequent subaerial debris flow event. In any case, also df turbidites resulting from failure of the active fan delta slope are genetically linked to subaerial debris-flow activity, even if they may not always correspond to an actual singular subaerial debris flow event in a one-to-one manner, but may represent delayed subaqueous fan delta slope failure. Even if we are potentially misinterpreting one or the other df turbidite that still might be related to some other unknown trigger, such few wrongly interpreted turbidites have no major impact on the inferred debris-flow frequency or the interpretation on intense precipitation as we apply 21-yr and 150-yr bandwidths for the frequency analyses."

**(1) The robustness of the distinction of 'df turbidites' and 'eq turbidites' based on geochemical and grain-size profiles**

Line 715 of the Conclusion states:
"Df turbidites show a graded grain-size trend, have a less steep D50/D90 ratio and contain more TOC compared to the homogeneous eq turbidites. ,,
In my mind this statement is not fully supported by the data shown in Figure 6.

The two distinctly looking grain-size profiles of Fig 6a obviously do not prove that there is a statistical difference between the two types (if there are only 2 types of tubiidites, see my doubts above…) in terms of the grain-size grading of the deposit. The authors need to show that there is a robust difference in the grading of the two deposit-groups based on the analyses of many deposits.

In my view Fig. 6b shows no significant difference between the different turbidites types in TOC-d13C space. If the authors want to state a difference here, they need to show by e.g., a cluster analysis, that these clusters are actually statistically distinct and the two groups cannot be drawn from a single distribution. How does the sample size (many df , fewer eq and background data points) affect the analyses? The same applies to the data shown in figure 6c: Is the difference in D50/D90 space significant? How would acquiring more eq data points affect the result?

(2) The use of grain-size trends and slope of D50-D90 evolution and their significance for event distinction is supported by several studies in different lacustrine environments (see e.g. Wilhelm et al., 2017; Vandekerkhove et al., 2020, and references therein). It is also important to note that these data points are not arbitrary sampled from the deposits, but do represent closely spaced samples of the whole deposits and thus represent the grain-size trend of the whole deposit (see e.g. Beck 2009). However, besides this reasoning, we will never be able to show a statistically significant distinction as asked by the reviewer, because there are just too few earthquake-triggered turbidites to obtain statistically relevant results.

We agree with the reviewer that the TOC-$\delta^{13}$C values are not statistically significant from each other. However, it makes totally sense that the background sediment in a lake like Plansee dominated by mixed hemipelagic-clastic influenced sedimentation also shows a mixed organic matter signature of terrestrial (C/N ratio >20) and limnic (algal) organic matter (C/N ratio <10) (Meyers and Teranes, 2001). This further also applies to the df-turbidites (pure terrestrial OM) and eq turbidites (remobilization of mixed hemipelagic-clastic slopes and thus the least OM). We do not consider the TOC-$\delta^{13}$C values as the most important proxy for event type distinction, but we use it in a difficult (mixed hemipelagic-clastic) setting as another small hint to make interpretations on the underlying process (Lines 420–422; 447–454; 460–461).

Wilhelm, B., Vogel, H., and Anselmetti, F. S.: A multi-centennial record of past floods and earthquakes in Valle d'Aosta, Mediterranean Italian Alps, Nat. Hazards Earth Syst. Sci., 17, 613–625, https://doi.org/10.5194/nhess-17-613-2017, 2017.

Vandekerkhove, E., Van Daele, M., Praet, N., Cnudde, V., Haeussler, P.J. and De Batist, M.: Flood-triggered versus earthquake-triggered turbidites: A sedimentological study in clastic lake sediments (Eklutna Lake, Alaska), Sedimentology, 67, 364-389. https://doi.org/10.1111/sed.12646, 2020.

Beck, C.: Late Quaternary lacustrine paleo-seismic archives in north-western Alps: Examples of earthquake-origin assessment of sedimentary disturbances''. Earth-Science Rev., 96, 327–344, 2009.

Meyers, P. A. and Teranes, J. L.: Sediment Organic Matter, in: Tracking environmental change using lake sediments, edited by: Last, W. M. and Smol, J. P., Kluwer Academic Publishers, Dordrecht, 239–269, https://doi.org/10.1007/0-306-47670-3_9, 2001.

**Age & turbidite thickness & frequency uncertainties**

(1) Unclear how the 137Cs & 210Pb data informed the BACON model in Fig S2. Did it?

(2) We mainly used the $^{137}$Cs peaks related to the Chernobyl nuclear accident in 1986 and the nuclear bomb tests in the bikini atoll in 1963, as outlined in Table S2. For a better clarification, we also revised the figure caption in the Supplement Figure S2.

(3)

>Supplement Lines 8–11

"Fig. S2: Age-depth model of the most distal core Plan 18-10 (recalibrated after Oswald et al., 2021), established by Bayesian age–depth modelling of radiocarbon ages (dark blue) and short-lived radionuclide dates (light blue) using the R-software Bacon v 2.4.3. Horizontal dashed line depicts the stratigraphic boundary between Phase 2 and Phase 3 and facilitates the age-depth model to abruptly change the sedimentation rate if required by the dates."

(1) The authors address the uncertainties in the age models by showing estimates for every single BACON simulation, a valid approach in my mind. However, this approach is only shown for the 150 yr bandwith, not for the 21 yrs. Hence, the uncertainties on this model cannot be assessed. Please add.

(2) We addressed this comment by adding an additional supplementary figure, where the single age simulations of the 21-yr frequency are shown (see Supplement Fig. S3).

(1) Can you comment on the uncertainties on (often very thin) turbidite thickness measurements due to the often diffuse upper boundary to the background sedimentation?

(2) Df turbidites are in general sharp bound units with macroscopic sedimentological characteristics (e.g. coarse base, distinct clay cap) and are therefore also well mappable within 600 dpi core pictures following the sedimentological criteria outlined in section 3.3.3.

(1) Moreover Table 1 shows a bunch of parameters for each 'phase' with no uncertainty estimates. These need to be added to make this comparison between 'phases' meaningful.

(2) We addressed this comment by adding uncertainty values for the recurrence interval and deposition rate to Table 1 based on the minimum and maximum ages derived from the age-depth model.

(1) Difference in Phase 3 and 4 Fig 8a: Is the increase in cumulative thickness between phase 3 and 4 actually significant when the uncertainties are accounted for? Cannot tell from this plot.

(2) Yes this change is significant as it is derived by a statistical test i.e. change-point analysis as outlined in Lines 277 and 502–504 in the revised manuscript.

(3)

>Lines 502–504

"Strikingly, the autonomously differentiated df phases using a change point analysis on the interevent times of df turbidites temporally coincide with the above-described lithotypes indicating the major influence of debris flows to subaquatic event deposition in Plansee."

(1) Line 695 and ongoing:
„we infer a mean frequency of phase 4.1 and 4.2 to be a best estimate, showing a ~7-fold increase in phase 4.2 compared to phase 3 (~1520 to ~1920 CE) coincident with the instrumentally documented increase in rainstorm activity"
Interesting, but I cannot follow this argument as the rainfall data are not shown in your figures as only

the precipitation record of the past 118 years is shown in figure 8b. This is an important observation, to convince your reader you need to show the data & the robustness of the correlation.

(2) We refer to daily precipitation sums of the nearest meteorological station "Berwang", which have been analyzed by Dietrich and Krautblatter (2017). The frequency of heavy rainfall events ($\geq$35 mm d$^{-1}$) has doubled from 1920 to 2010, with an average increase of 10 % per decade (Hydrographischer Dienst Tirol, 2020). We introduce these results in Line 86 and we added the information to the addressed sentence.

Hydrographischer Dienst Tirol (eHYD): https://ehyd.gv.at/, last access: 7 January 2021.

Dietrich, A. and Krautblatter, M.: Evidence for enhanced debris-flow activity in the Northern Calcareous Alps since the 1980s (Plansee, Austria), Geomorphology, 287, 144–158, https://doi.org/10.1016/j.geomorph.2016.01.013, 2017.

(3)

>Lines 741–744

"Without the possibility of quantifying these human influences, we infer a mean frequency of phase 4.1 and 4.2 to be a best estimate, showing a ~7-fold increase in phase 4.2 compared to phase 3 (~1520 to ~1920 CE) coincident with the instrumentally documented 2-fold increase in rainstorm activity ($\geq$35 mm d$^{-1}$) from 1920 to 2010 (cf. Dietrich and Krautblatter, 2017)."

**Sadler effect-related issues regarding the increase of turbidite frequency and thickness**

(1) Turbidite thickness and deposition rate is generally increasing towards younger deposits. Can these changes be simply attributed to the statistical bias describes as the Sadler effect as older strata is more prone to hiatuses and sediment removal (Sadler, 1981; Sadler and Jerolmack, 2014).

(2) Since lake Plansee is a nearly closed system with virtually no sediment removal or erosional processes in the deep main basin, the Sadler effect seems to be irrelevant. The core from the deepest location was chosen for dating of the sediment succession, where the age-depth model shows no sign of a hiatus. In addition, we corrected the sediment depth for water content to account for sediment compaction during sampling. In this way, we improved the comparability of the debris flow thicknesses (and slope of the cumulative thickness curve) between the recent deposits of phase 4 and older sequences.

**The value of the sediment delivery ratio in this context**

(1) Line 188, Terrestrial LIDAR data Equation 1
I do not understand how you determine the numerator: sediment flux into the lake, by 3 LiDAR surveys which are 3 month apart. What about the sediment flux from farther up the catchment that bypassed the survey area and ended up in the lake? Please clarify.
Line 395 states: During the 3 months, the sediment only accumulated on the terrestrial fan delta. How do you know? Differential bathymetry on the lake bottom? Sediment traps employed? How do you now no sediment bypassed into the lake?
I guess your argument is the sediment delivery ratio. But (if I understand this correctly) that ratio is only based on data of the fan itself. However, material mobilized further up in the catchment could have simply bypassed the fan (eroded some, deposited some on the alluvial fan) and deposited in the lake. I do not think this ratio is useful to determine how much sediment made it into the lake during the 3-month period, if not the entire catchment draining into the lake is imaged & considered.

And one more:

Line 577 I am not sure you can use the sediment delivery ratio – the way you calculated this – as a measure for system connectivity, as there might be much more sediment involved from further up in the catchment making it into the that is not accounted for by your difference maps of the fan. I don't think this does much to your story. Would just leave this out and shorten the manuscript somewhat.

(2) We clarified in our manuscript how we calculated the sediment flux (difference between erosion and deposition volume) into the lake (Line 194). We are aware that the sediment delivery ratio is only calculated for the LiDAR-covered area since we are not able to cover the upper catchment area in non-accessible, steep terrain. Our premise derived from field observation, now indicated in the article (Lines 178–180; 564–566), is that the LiDAR covered area of the fan includes the major proportion of redistributed sediments since the higher inaccessible steep catchments cannot accumulate significant amounts of debris. Furthermore, the mature fan is below the deposition angle of debris (<15°; Grelle et al., 2019) which necessarily forces smaller debris flows to deposit a major share of their initial volume. Thus, we just use the sediment delivery ratio as an indicator for the proportional sedimentation on land and in the lake; this is to demonstrate that there is a certain proportionality of sedimentation respective to age and steepness of fans. We added this point in our discussion (Lines 631–634).

Grelle, G., Rossi, A., Revellino, P., Guerriero, L., Guadagno, F. M., & Sappa, G.: Assessment of Debris-Flow Erosion and Deposit Areas by Morphometric Analysis and a GIS-Based Simplified Procedure: A Case Study of Paupisi in the Southern Apennines. Sustainability, 11(8), 2382. https://www.mdpi.com/2071-1050/11/8/2382, 2019.

(3)

>Lines 178–180

"Our premise derived from field observation is that the LiDAR covered area of the fan includes the major proportion of redistributed sediments since the higher inaccessible steep catchment parts cannot accumulate significant amounts of debris."

>Lines 192–194

"The Sediment Delivery Ratio (SDR) is an indicator for the proportional sedimentation on land and in the lake. It was calculated following Eq. (1) (Lu et al., 2006), where the sediment flux into the lake is the difference between erosion and deposition volume."

>Lines 406–407

"In total, 41 m$^3$ of debris were eroded, and 651 m$^3$ were deposited onshore, while no sediment flux into the lake was detected in the investigated period."

>Lines 564–566

"The upper, inaccessible catchment parts could not be covered by LiDAR, but based on field observations we infer that they cannot accumulate significant amounts of debris and the sediment redistribution is concentrated in the LiDAR-covered channel and delta area."

>Lines 631–634

"The mature fan delta mainly experienced terrestrial deposition during the investigated period. The average 10° slope is below the deposition angle of debris (<15°, Grelle et al., 2019), which necessarily forces smaller debris flows to deposit a major share of their initial volume. Comparison of the sediment delivery ratio of the two fan deltas demonstrates that there is a certain proportionality of on- and offshore sedimentation respective to the age and steepness of the fans."

**Landscape recovery from earthquakes**

(1) This is a very interesting aspect that may deserve more attention in some other manuscript. However, if I read figure 7a correctly – events during the "P-period' can only be recognized in the proximal cores Pan 19-03 and Plan19-04 because the two distal cores (Plan18-10 and Plan 19-02) do not penetrate deep enough to even capture these events. So how do you know these events are not recorded in these locations but were simply note penetrated by the two distal cores?

(2) One of the two distal cores reached deep enough in the stratigraphy to have recorded both the eq turbidite and two turbidites interpreted as the postseismic landscape response. The small turbidites from the "P-period" did not reach this distal core. For a better visualization of our core-to-core interpretation, we increased the line thickness in Figure 7.

Minor comments:

(1) It sure is a challenge to derive useful information of datasets covering completely different time scales. The 'amphibious' approach is much praised in the intro, however, what are exactly the new insights that we gain form this approach? This falls somewhat short.
In my view you can show that the currently active parts of the alluvial fan-delta system, that active delta front is dominated by debris flow lobes and erosion and deposition happens on the alluvial fan.

(2) We conduct on- and offshore investigations to find evidence for the processes governing the lake sedimentary system and to ascribe a single flow type to deposits in the sediment cores. Instead of inferring the process indirectly from the deep-basin sediment record on its own, we investigate the geomorphic fan evolution in order to understand the event stratigraphy. With the amphibious approach we (i) uncover conclusive evidence of subaquatic debris-flow deposits and subaquatic landslide deposits, which enables us to pinpoint the detrital layers to single debris-flow events and subaquatic slope failures in the sediment cores (Fig. 3); (ii) we justify the transect and coring site selection by documenting subaquatic expressions on different fan delta types (Line 220) (iii) we link wedge-shaped deposits throughout the 4,000 year sediment profile to low-energy debris flow-activity on the juvenile fan delta (Line 654); and (iv) we correlate subaquatic deposits showing high backscatter signals with previously mapped terrestrial debris-flow deposits from 1947 to 2014 (Fig. 2).

(1) In line 98, the authors state that there is no permanent river inflow to the main basin of the Plansee. However, Line 112 mentions two permanent inflows /draining catchments) into the lake. Please clarify the discrepancy. By main basin you mean your specific study area? And these basins are well separated by a high morphologic barrier in between that turbidity currents cannot cross? Does not look like it in your figure 1…

(2) The two permanent inflows are located in the NE and E of the lake (black arrows Fig 1). The deposits from the large alluvial fan in the center of the lake reach far into the lake towards the slopes from the northern shoreline, creating a narrow and morphologically elevated passage (20 m elevated with respect to the eastern basin) between the eastern and the western part of the lake. This acts as a barrier for potential underflow turbidity currents originating from the two small streams.

(1) Fig. 3 and line 590
What do you mean by overspilling debris flows. Spilling over what? Beyond the break of slope onto the lake floor

(2) Thank you for pointing out the lack of clarity. We describe debris flows spilling over the currently active channel levee changing the original flow direction. We adapted our explanation in the manuscript.

(3)

>Lines 341–342

"Subaquatic debris flow-induced turbidites (df turbidites) along diffuse inactive channels are interpreted as debris-flow events spilling over the currently active channel levee."

>Lines 635–638

"However, the massive subaquatic volume of the fan delta and debris-flow deposits spilling over the currently active channel levee show that over time, most of the debris-flow material accumulates in the lake basin, and the onshore fan thickness represents only part of the total transported volume."

(1) Line 119 states that
„While the human interference may influence the continuous background sedimentation in the lake quite a bit, the episodic debris flows eroding materials from 10,000 m2 large catchments way above the lake will not be relevantly influenced." How are you so sure? Is there no human activity at all in the catchments above the lake? Deforestation during historic times/ reforestation after etc?

(2) Human activity is restricted to the shoreline of lake Plansee, where artificial lake level changes, wave action, and construction works impact the coastal erosion (Lines 692–694). The vegetation cover and succession in the catchment area is mostly controlled by the debris-flow activity (Dietrich & Krautblatter, 2017). The Plansee catchment represents a near-pristine area with respect to human activity due to high sediment redistribution, poor soils unsuited for mountain pasture, steep forested slopes (formerly used as royal hunting ground) and inaccessible, partially non-vegetated upper catchment areas (Hibler, 1921). Pollen records from the nearby Heiterwanger See reveal local forest clearance in the centuries around the beginning of the Common Era and during medieval times (Oeggl, 2004; Kral, 1989). The deposits in the sediment cores from Plansee show no sudden increase in debris flow frequency during these periods of forest use. No marked increased forest clearance is reported since medieval times (see Lines 684–689). We therefore infer that the immediate debris-flow prone areas were not affected by anthropogenic forest clearance.

Dietrich, A. and Krautblatter, M.: Evidence for enhanced debris-flow activity in the Northern Calcareous Alps since the 1980s (Plansee, Austria), Geomorphology, 287, 144–158, https://doi.org/10.1016/j.geomorph.2016.01.013, 2017.

Hibler, I. J.: Der Plansee und seine Umgebung, Universitätsverlag Wagner, Innsbruck, 135 pp., 1921.

Oeggl, K.: Palynologische Untersuchungen zur vor- und frühgeschichtlichen Erschließung des Lermooser Beckens in Tirol – Berichte der Reinhold-Tüxen-Gesellschaft – 16: 75–86, 2004.

Kral, F.: Pollenanalytische Untersuchungen im Fernpaßgebiet (Tirol): Zur Frage des Reliktcharakters der Bergsturz-Kiefernwälder. Verhandlungen der Zoologisch-Botanischen Gesellschaft von Österreich 126: 127–138, 1989.

(1) Line 300 and ongoing.
It seems like the authors assume that debris flows are the only sedimentary process happening on an alluvial fan. What's your evidence for that? Are there no fluvial or sheet flow processes happening? 100% debris flow? Are these deeply enough incised (4000 yrs of stratigraphy) and these outcrops show only debris flow deposits and no fluvial processes?

(2) We addressed this comment by adding a paragraph to the sections 2 study site, 3.3.3 and 5.1 discussion. We now better i) explain the peculiar setting at Plansee, ii) outline the two-fold event categorization as initial interpretation to be tested within this study and iii) discuss and justify the event categorization. For a detailed reply to this comment, we kindly refer to our reply to the main comment of this referee.

(1) Line 410

The Lightness L* factor of the sediment is first mentioned here but should be explained in the Methods section already

(2) We addressed this comment by adding the information to the methods section (3.3.2).

(3)

>Lines 223–224

"A core-to-core stratigraphic correlation was conducted based on sedimentary facies, density, lightness (L*) and distinct marker layers."

(1) Avoid unnecessary repetition of words throughout the manuscript, e.g.,
Line 42 two times "worldwide"
Line 16 and 18: "amphibious" (is this a good word for what you actually mean – combined terrestrial and lacustrine data acquisition)

(2) Thank you for pointing this out, we deleted the repetition in the manuscript. We believe the word "amphibious" describes our approach fittingly, as there are also 'amphibious' IODP-ICDP projects https://www.icdp-online.org/fileadmin/icdp/media/doc/2019_ICDP_FlyerEcord2019.pdf.

(1) Figure 2d and others.
To ensure inclusiveness, make sure your figures are legible to people with certain color-blindness ( 8% of all men and 0.5% of all women are colorblind). I am not colorblind (I think ;) but the differences in color are hard for me to see. I used this website to check your Fig 2d
https://www.color-blindness.com/coblis-color-blindness-simulator/
and apparently the figure looks like this for people with Red-Blind/Protanopia & your differently mapped areas are undisguisable.
Enhance the contrast between the colors and this will work better.

(2) Thank you for hinting at this issue. We will get in contact and ask the handling editor if and which figures need to be revised.

**Major remark**

The temporal analysis of occurrence rates is neither robust nor valid in the present form of the manuscript. Any analysis of the frequency spectrum of events in lake archives is essentially an analysis of the underlaying age-depth function. As I understand it, you are using the mean (or median) scenario from Bacon, which is constrained by seven age estimates for the last 4000 years (see Figure S2). The corresponding sampling rate of one age estimate per 500 years cannot resolve potential rapid sedimentation changes in this record. This sampling rate restricts the analysis to the millennial frequency band (see Nyquist-Shannon sampling theorem). Changes in higher frequency bands (as analyzed in the 21-yr window, see line 255) cannot be resolved. As a result, the inferred changes in occurrence rates during period 1 to 4 might be pure artefacts of the sampling strategy.

Obviously, an analysis of debris flows in the millennial frequency band is not satisfying, especially considering the detailed record at hand. From my point of view, there are two viable options for increasing the resolution of your analysis:

- Increase the sampling rate for the mean age-depth function. An analysis in decadal resolution requires at least two age estimates per decade. Varve counting could be a viable option if the finely laminated sediments in your record represent annual laminae.
- Making use of age uncertainty. Blaauw et al. (2018) showed, that a sample rate of two samples per millennia (as in this case) stabilizes the precision of Bayesian age-depth models. This suggests that individual simulations of Bacon can mimic higher-frequency changes in sedimentation patterns. Repeating the frequency analysis for different Bacon model

simulations in a Monte-Carlo approach could therefore be used to decrease the bandwidth and construct confidence limits on the occurrence rates. Individual model simulations of Bacon are readily accessible in the output of the algorithm (with accumulation rates (years per spacing) usually stored as .out file in the default folder). I am not sure how much this approach would allow to decrease the bandwidth, so I suggest coupling your analysis with a suitable bandwidth selection test (see e.g. Muddelsee, 2014 or Merz et al., 2016 and references therein).

There might be other approaches, but the design of the frequency analysis in its current form requires major revisions to make inferences on debris flow occurrence robust and bring it in line with the excellent set up of this study.

(2) We thank the referee for this important remark along with providing potential options to improve our frequency calculations. We fully agree with the referee, that the presentation of the occurrence rate of df turbidites solely based on their mean ages was imprecise in the previous manuscript, as the age model uncertainties were neglected. In the revised manuscript we have now followed the referee´s recommendation to apply the frequency analysis on a data-based bandwidth coupled with the presentation of the age uncertainty also in the occurrence rate plot as provided by the individual simulations of the age–depth modelling software Bacon (see more details in the specific reply comment below). With this, we think, that the revised version of the manuscript could be greatly improved and now addresses the main critical comment by the referee. However, we do not fully agree with the referee, that in a lacustrine setting the bandwidth selection must follow the Nyquist–Shannon sampling theorem. Lacustrine sedimentation in deep open lakes contain the important characteristic that they are often and in general continuous. In turn, continuous sedimentation adds confidence in reducing the bandwidth for frequency analyses, which has also been done in the past by a similar amount of age constraints and number of event deposits (see e.g. Wirth et al. 2013 and references therein).

In Plansee there are no indications that support drastic changes in sedimentation rate:

- Especially in such finely-laminated, mixed hemipelagic-clastic sediments as present in the sedimentary archive of Plansee, abrupt shifts in sedimentation also cause changes of the **lithotype**. Looking at the core images in Figure 7A, the only striking lithotype change is at the phase 2–phase 3 boundary. As this stratigraphic level is well dated with a radiocarbon date immediately (2 cm) below, we implemented it now as a stratigraphic "boundary" for the age depth modelling in Bacon, facilitating the model to change sedimentation rates if required given the distribution of the radiocarbon dates that inform the model (see revised supplement Figure S2). Therefore, also some of the reported ages slightly changed compared to the previous manuscript especially between 1920–1700 CE.
- Accompanying a change in lithotype, abrupt shifts in sedimentation rate in a clastic lake like Plansee would also cause a thickness increase of laminae, which is not observable in the Plansee sediments.
- Furthermore, abrupt shifts in sedimentation rate caused by phases of enhanced event deposition can be neglected, as all event deposits >5 mm are cut out from the age–depth model resulting in an event-free sediment depth, as also outlined in Line 249 in the revised manuscript.

We highly appreciate the potential options provided by the referee. Varve counting is not an option for this manuscript, because given the high amount of event deposits, varve counting may severely underestimate the ages. Imagine each deposit eroded 1 mm, meaning that with a total of ~138 events and a mean sedimentation rate of 0.45 mm/yr about 307 years of sediment are gone. Such potential minor erosion would artificially enhance peaks in the accumulation curve. Moreover, varve counting

would lead to a completely different manuscript and would distract from the main focus of this work, as we aim to trace the major debris flow events from onshore to offshore in a multidisciplinary way. Due to the oxic condition at the bottom water, we would also not expect to have a high preservation potential of varves. However, if the laminae are varves, these are most likely mixed biogenic–clastic varves, where a fundamental part of the varve is built up by the annually clastic input during the spring and summer months (basically the df turbidites). As we aim to study frequency of df turbidites, also using these as an age-constraint would lead to a circular reasoning.

Therefore, as also suggested by the referee, we made use of the age uncertainty provided by the age-modelling software Bacon and coupled this with a suitable bandwidth selection test. First, we applied a bandwidth selection test based on the number of df turbidites after Sheather and Jones (1991), which is also widely used e.g. for kernel statistics and often recommended in data science. This results in a bandwidth of 150.2 years for the whole 4,000 year record and in 16.5 years for the last two centuries, where df frequency is clearly enhanced. Second, we applied the frequency analysis using this 150-yr bandwidth on the event ages of each of the 6,396 individual age-model simulations derived from Bacon. The frequency analysis (with 150-yr bandwidth) on all simulations results in a 95% uncertainty belt providing a general overview of df turbidite frequency over time (Figure 8a), but likely underestimates/smoothens the actual frequency changes where more frequent events occurred, e.g. in the last two centuries. To overcome this issue, we coupled the 150-yr bandwidth occurrence rate diagram with the 21-yr frequency curve of the mean age to also provide a more detailed view on the frequency changes, especially where a higher number of events are present (e.g. the last two centuries). While this 21-yr frequency might be oversampled for the whole 4,000 year record, enough age constraints in the last century (1 coring date, 2 radionuclide ages, 1930 earthquake-induced turbidite) and high event frequency allows for more detailed bandwidth in the frequency analyses. Although we now provide the 21-yr and the 150-yr frequency curves for the whole record, we only discuss the detailed 21-yr frequency changes for the last century in more detail (see Lines 526–530).

The general frequency changes are also represented in the 150-yr frequency plot i.e. increased frequencies after the ~2120 BCE earthquake and for the last century (see Figure 8a), which further supports the interpretation of actual increased df activity at these times. We explain this approach in the methodology section in Lines 271–280 and in the results section in Lines 496–498 in the revised manuscript. Furthermore, we also added the age uncertainty range to the cumulative thickness curve (Figure 8a, Lines 280–281) and to the df turbidite table in the supplement (Table S4). However, we kept the mean ages for the thickness plots, as introducing age uncertainties would make this bar plot unreadable.

Wirth, S. B., Glur, L., Gilli, A., and Anselmetti, F. S.: Holocene flood frequency across the Central Alps - solar forcing and evidence for variations in North Atlantic atmospheric circulation. Quat. Sci. Rev. 80, 112–128. doi:10.1016/j.quascirev.2013.09.002, 2013.

Sheather, S. J., & Jones, M. C.: A reliable data-based bandwidth selection method for kernel density estimation. Journal of the Royal Statistical Society: Series B (Methodological), 53(3), 683–690, 1991.

(3)

>Line 249
"For the age–depth modelling, event deposits >5 mm were removed to obtain an event-free sediment depth."

>Lines 271–281
"Additionally, we calculated the annual occurrence rate of df turbidites using a central running sum with different bandwidths to reconstruct changes in debris-flow frequency over time in different resolutions. First, a suitable bandwidth (150 yrs) was selected based on the average df-turbidite

occurrence over the entire core (Sheather and Jones, 1991). We applied this bandwidth on the occurrence rate calculation for each individual simulation of the age–depth model derived from the R-software Bacon v 2.4.3 (Blaauw and Christen, 2011). This results in a data-based frequency analysis that incorporates age–depth model uncertainties. The rather broad bandwidth is suited for showing general changes in frequency over time. To account for a higher resolution in frequency changes especially in periods with higher number of events, we also calculated a bandwidth based on the df occurrence of the last two centuries and applied the resulting 21-yr bandwidth to occurrence rate calculation of the main age. The cumulative thickness over time involves both the thickness and frequency of df turbidites and its slope provides information on df turbidite accumulation rate per year. We calculated the cumulative thickness on the mean values and the 95 % range values of the age–depth model to transfer age uncertainty to the cumulative thickness analysis."

>Lines 496–498
"The frequency analysis of the whole 4,000 year df turbidite record is based on a 150-yr bandwidth, whereas especially the higher number of events in the 20th century requires higher resolution frequency analysis, here on the basis of a 21-yr bandwidth (Fig. 8a)."

>Lines 526–530
"Df phase 4.1 is represented by a strong and fast frequency increase at ~1920 CE, followed by a period of highly frequent debris-flow events ~1980 CE. Since then, the current df phase 4.2 has lower frequencies relative to phase 4.1 but still by far higher frequencies than in the main df phases 1–3. Debris-flow frequency in 4.1 increased by a factor of 8 compared to the reference df phase 3. In df phase 4.2, debris-flow frequency increased by a factor of 7 compared to df phase 3."

**Minor comments**

(1) Line 255 to 257: Please add a detailed description of the occurrence rate estimation once the approach has been adapted.

(2) We addressed this comment by adding a detailed explanation on our approach in the methodology section in Lines 271–280. Please see the reply to the major remark above for more details to the revised approach, that follows the referee's suggestion.

(1) Line 489–490: I can´t detect any changes in the mean thickness of layers during phase 3. The cumulative function stays linear in my opinion. Can you quantify the change?

(2) We thank the referee for hinting at this flaw. The mean thickness of debris-flow deposits increases in the second half of df phase 3, but we cannot deduce exponential growth. We deleted this sentence and now only describe the frequency change between the phases, where the increase can be quantified by the change of accumulation rate of df turbidites from 0.1 mm/a (df phase 2) to 0.2 mm/a (df phase 3) as outlined in the revised manuscript in Lines 521–522 and also noted in Figure 8a.

(1) Line 574: Consider replacing "prove" with "suggest".

(2) We thank the referee for the suggestion. We revised to "suggest" in Line 593.

(3)

>Lines 593–594
„Overprinted lobes of few large debris flows on the active subaquatic fan area suggest that terrestrial deposition dominates recently."

(1) Line 618–624: Has the vegetation cover stayed constant during the last 4,000 years? Are there pollen records from Lake Plansee or different archives in the Eastern Alps which back up your line of argument?

(2) We agree with the referee that further information from pollen is needed to substantiate our statement. The next pollen record comes from Heiterwanger See a few kilometres west of Plansee. The pollen record reveals first signs of permanent settlement during the Iron Age. Local forest clearance in the centuries around the beginning of the Common Era created arable land and resulted in vegetation changes (Oeggl 2004, Kral 1989). The following Migration Period is characterized by little human activity and reforestation. A period of enhanced forest clearance occurred during medieval times. Agriculture and grazing played a larger role then compared to today. The deposits in the sediment core from lake Plansee show no sudden increase in debris flow frequency during this period of intense forest use. Pollen diagrams from an adjacent Alpine area (Ostallgäu) let us infer that in the Plansee area, forest clearance and settlement decreased during the Little Ice Age (Stojakowits & Friedmann, 2013). No marked increased forest clearance is reported since medieval times and we added this information to the respective paragraph (Lines 635–644).

Kral, F.: Pollenanalytische Untersuchungen im Fernpaßgebiet (Tirol): Zur Frage des Reliktcharakters der Bergsturz-Kiefernwälder. Verhandlungen der Zoologisch-Botanischen Gesellschaft von Österreich 126: 127–138, 1989.

Oeggl, K.: Palynologische Untersuchungen zur vor- und frühgeschichtlichen Erschließung des Lermooser Beckens in Tirol – Berichte der Reinhold-Tüxen-Gesellschaft – 16: 75–86, 2004.

Stojakowits, P. and Friedmann, A.: Pollenanalytische Rekonstruktion der Vegetations- und Landnutzungsgeschichte des südlichen Ostallgäus (Bayern). In: TELMA - Berichte der Deutschen Gesellschaft für Moor- und Torfkunde, Band 43, 55–82, DOI 10.23689/fidgeo-2867, 2013.

(3)

>Lines 635–644
"Human-induced vegetation changes are documented since about 1,000 BCE from a pollen record from peat bog remnants at Heiterwanger See a few kilometres west of Plansee (Kral, 1989). A period of enhanced forest clearance in the area happened during medieval times according to this record. The herein presented sedimentary record of Plansee shows no signs of a drastic increase in debris-flow frequency throughout this period of intense forest use. During late medieval times, the climatic deterioration of the Little Ice Age, war, and epidemics led to a decrease in population and in forest clearance, which can be observed in pollen diagrams from the nearby Ostallgäu (Stojakowits and Friedmann, 2013). Since then, no further increase of forest clearance or wildfires is reported in the area. Therefore, we infer that there were no significant changes of vegetation before or during the period of increasing debris-flow frequency in the last century (df phase 4)."

(1) Line 647: How can you be sure, that both layers correspond to the heavy rainstorms in 1999 and 2005, especially considering age uncertainty and that not every df turbidite has a corresponding rainstorm event (see lines 652-653)?

(2) We have addressed this comment and added more clarity on the uncertainty by adding "potential temporal overlap" to the sentence in Line 676 and by stating the age uncertainty (Lines 684–685).

(3)

"Potential temporal overlap of two df turbidites with the two heaviest rainstorms of the century in 1999 and 2005 CE (Fig. 8b) let us infer that regional (advective) rainstorms lasting over several days are a trigger mechanism of debris flows also in the Plansee region."

"Potential mismatches might also be due to the age error of df turbidites, which has a 95 % probability range from 5 years for the youngest event to 60 years at the begin of df phase 4."

(1) 649-651 You stated in line 540-541, that you would refrain from detailed comparisons to other flood deposits.

(2) We refrain from a detailed (statistical) comparison to other flood deposits for discussing potential climatic drivers on debris-flow activity, because of the greater age uncertainties in our df turbidite record. However, we do not consider the qualitative comparison of several outstanding thick df turbidites in the first and eighth century BCE to enhanced flood activities from other archives as detailed.

(1) Line 661-664 This passage is difficult to relate to passage iv (line 625-634), in which you argue that human influence enhanced coastal erosion and subsequently the volume (not the frequency) of debris flows. If it is only influencing the volume of debris flows, how can human influence affect the frequency of debris flows recorded in the core?

(2) We clarified the possibility that some of the df turbidites in the first of half of the $20^{th}$ century might reflect human-induced mass wasting events in passage iv (Lines 649–650).

(3)

"Therefore, some of the df turbidites in the first half of the 20th century might actually reflect human-induced mass wasting (see also vi)."

(1) Line 666-667: Does "we" refer to this study or to the results of Diedrich and Krautblatter, 2017?

(2) We increased the clarity by adding "as previously observed" to the sentence in Line 698.

(3)

"Moreover, we herewith provide sedimentological evidence for the increased debris-flow activity in the 20th century, as previously observed on differential LiDAR data from several fans at the northern shore of the lake (Dietrich and Krautblatter, 2017)."

(1) Line 673-676: The absence of other increases in occurrence rates might be an artefact of the applied methodology (see major remark).

(2) We kept the interpretation that debris flow activity is strongly controlled by local high-intensity convective precipitation, as also after applying the frequency calculations on the appropriate bandwidth (150 years) there are no larger frequency changes except the one for the $20^{th}$ century and after the ~2120 BCE earthquake (Lines 718–720).

(1) Figure 4: The main channel in August is difficult to distinguish from the red background on the map.

(2) We have changed the color of the main channel outline for better visibility.

(1) Figure S2: Age units correspond to BP in this figure, in contrast to BCE/CE notation in the rest of the manuscript.

(2) We have changed the notation to BCE/CE in the new Figure S2.

(1) Table S2: This table only contains five 14C measurements, contrary to six radiocarbon dates shown in Figure S2.

(2) We thank the referee for pointing out this mistake. We have added the other radiocarbon date to Table S2.

We thank the referee for the constructive and insightful comments which we feel helped to improve the manuscript. We will upload a revised version of the manuscript and below provide a point-by-point response to the comments. Our response is structured as follows: (1) referee comment, (2) author's response and (3) changes in manuscript.

Many thanks and best regards,

Carolin Kiefer, on behalf of all authors

**RC2: 'Comment on esurf-2021-23', Anonymous Referee #2, 28 May 2021**

General comments

The submitted study shows the timing over 4000 years of debris flow events from transport-limited catchments, much of whose deposits extend into an adjacent Lake, creating subaqueous geomorphic field evidence by lacustric deposits. The knowledge of such a long time series is, besides geomorphological interest, also important in connection with positive trends in climate change, or land use, exposure, etc. and gives valuable insight to recent dynamics in the occurrence of debris flow events. The article is basically well written and understandable. The literature cited is up to date and, and here I focus exclusively on the presentation of an extraordinary time series of debris flow events, an important and pioneering contribution to the research community.

(1) Precisely because, at least to me, no comparably long historical review of debris flow events is known, a publication of such a time series requires precise information on uncertainties due to the applied methodology and a discussion of how to deal with it.
Above all I recommend focusing on the creation and trend analysis of the surveyed frequency representation of such a long (and valuable) debris-flow time series, improving reliability of the time series.

(2) We thank the referee for this remark on the reliability of the debris-flow time series. We fully agree that calculating the age model uncertainties is a necessary improvement on the precision of our frequency calculations. In the revised manuscript, we present an improved age model, which now provides ages at 1 mm steps. In addition, we apply a frequency analysis on a data-based bandwidth coupled with the presentation of the age uncertainty also in the occurrence rate plot as provided by the individual simulations of the age-depth modelling software Bacon. For a more detailed answer and the changes in the manuscript, please view our answer to the major comment on uncertainties of age dating below.

(1) All terrestrial and bathymetric observations should be either shifted to a second paper or significantly shortened.

(2) We thank the referee for the suggestion. In the first version of our manuscript, we did not put enough emphasis on describing the necessary and previously underestimated link between on- and offshore morphological studies and the debris flow turbidite record. Finding evidence for the underlying processes was a prerequisite for defining identification criteria and understanding the event stratigraphy in the sediment cores. The geomorphic description of debris-flow fan evolution should therefore not be decoupled from the event deposit interpretation. We shortened the manuscript with regards to recent morphological changes (e.g. section 5.2) and put more emphasis on conclusive system understanding from a source-to-sink perspective. For a more elaborate explanation, please refer to the answer on the first specific major comment below.

Specific major comments

(1) The introduction to the study describes the need for knowledge of long-term time series, with the reader curious about this 4000-year survey. Interestingly, however, it talks about a combination of several geomorphological surveys in different depositional domains. First of all it must be noted that for the creation of the time series it is not comprehensible why on two selected fans a terrestrial and bathymetric evaluation is carried out, which is not related to the time series collected and analyzed due to lacustrine deposits at all. It is argued that the coupled study of debris flow systems on land and underwater will provide new insights into geomorphic expressions from the catchment to the depocenter provide? So what exactly are these new insights and how do they relate to a 4000 year time series? I was under the impression that it was more like two stories in one study. Whereby the terrestrial as well as bathymetric studies are mainly closer in the context of the already published article by Dietrich and Krautblatter (2017). By the way, since the onshore measurement is based on two LiDAR measurements within a year, the question arise how does one use this to elucidate the relationship between terrestrial and subaquatic deposition of recent debris flows? In other words, how and why should the rates differ from chance - based on the scarcity of data? My suggestion would therefore be that the authors refer, in the present article, more or exclusively to the survey of lake sediments and their statements in connection with debris flow frequency. Although the terrestrial and bathymetric investigation contains some interesting and further information, they seem to be dispensable for the creation of the 4000 years time series. It would be more exciting to focus on the time series of debris flows and find out how well you can determine the frequency based on sediment core analyses.

(2) The referee expresses concern on the link between the geomorphic study and the observations in the sediment cores. In this article, we establish a general understanding of how the major debris flows are transported to their final sink and how their deposits are distributed in the basin before interpreting the turbidite layers found in the 4,000 year time series. On- and offshore geomorphic investigations provided the following new insights: (i) we uncover conclusive evidence of subaquatic debris-flow deposits and subaquatic landslide deposits, which enables us to pinpoint the detrital layers to single debris-flow events and subaquatic slope failures in the sediment cores (Fig. 3); (ii) we justify the transect and coring site selection by documenting the recent debris-flow activity and subaquatic expressions on different fan delta types and their catchments (Lines 156-161); (iii) we link wedge-shaped deposits throughout the 4,000 year sediment profile to low-energy debris flow-activity on the juvenile fan delta (Lines 606-614); and (iv) we correlate subaquatic deposits showing high backscatter signals with previously mapped terrestrial debris-flow deposits from 1947 to 2014 (Fig. 2).

The referee doubts the representativeness of our onshore data due to the limited observation period of only three months. We agree on this statement, but by including the bathymetric data and further assessing the evolution of debris flow-deltas, we can understand the geomorphic system and implement the shorter recorded period into a longer framework. The bathymetric data confirmed active progradation on the juvenile fan delta and its significant contribution to the subaquatic sediment deposition, which was indicated by the TLS results. We therefore chose the juvenile fan as a starting point for the core transect. We changed the text (Lines 580-586) accordingly.

(3)

[revised manuscript text omitted]

(1) Concerning the lacustrine event deposits, I find the methodology for event type differentiation very exciting and comprehensible, also in the awareness that it is one of the rare opportunities to observe the process of debris flow in a relatively uninfluenced setting (which, by the way, is also well executed). Nevertheless, the following data processing raises questions.
My mainly concerns regarding this study relate primarily to the uncertainties of age dating. What uncertainties are there in the temporal determination? Which in the event detection? How do these affect the calculated frequency?

(2) We thank the referee for making an important remark on the age-model accuracy. In our new manuscript, we show a revised approach following comments of both referees, which we think greatly improved the age-depth model. We made use of the age uncertainty provided by the age-modelling software Bacon and coupled this with a suitable bandwidth selection test. First, we applied a bandwidth selection test based on the number of df turbidites after Sheather and Jones (1991), which is also widely used e.g. for kernel statistics and often recommended in data science. This results in a bandwidth of 150.2 years for the whole 4,000 year record and in 16.5 years for the last two centuries, where df frequency is clearly enhanced. Second, we applied the frequency analysis using this 150 years bandwidth on the event ages of each of the 6,396 individual age-model simulations derived from Bacon. The frequency analysis on all simulations results in a 95% uncertainty belt providing a general overview of df turbidite frequency over time (Figure 8a), but likely underestimates/smoothens the actual frequency changes where more frequent events occurred, e.g. in the last two centuries. To overcome this issue, we coupled the 150-yr bandwidth occurrence rate diagram with the 21-yr frequency curve of the mean age to also provide a more detailed view on the frequency changes, especially where a higher number of events are present (e.g. the last two centuries). While this 21-yr frequency might be oversampled for the whole 4,000 year record, enough age constraints in the last century (1 coring date, 2 radionuclide ages, 1930 earthquake-induced turbidite) and high event frequency allows for more detailed bandwidth in the frequency analyses. Although we now provide the 21-yr and the 150-yr frequency curves for the whole record, we only discuss the detailed 21-yr frequency changes for the last century in more detail (see Lines 526-530).
The general frequency changes are also represented in the 150-yr frequency plot i.e. increased frequencies after the ~2120 BCE earthquake and for the last century (see Figure 8a), which further supports the interpretation of actual increased df activity at these times. We explain this approach in the methodology section in Lines 271-280 and in the results section in Lines 496-498 in the revised manuscript. Furthermore, we also added the age uncertainty range to the cumulative thickness curve (Figure 8a, Lines 280-281) and to the df turbidite table in the supplement (Table S4). However, we kept the mean ages for the thickness plots, as introducing age uncertainties would make this bar plot unreadable.

Sheather, S. J., & Jones, M. C.: A reliable data-based bandwidth selection method for kernel density estimation. Journal of the Royal Statistical Society: Series B (Methodological), 53(3), 683-690, 1991.

(3)

>Lines 271-281
"Additionally, we calculated the annual occurrence rate of df turbidites using a central running sum with different bandwidths to reconstruct changes in debris-flow frequency over time in different resolutions. First, a suitable bandwidth (150 yrs) was selected based on the average df-turbidite occurrence over the entire core (Sheather and Jones, 1991). We applied this bandwidth on the occurrence rate calculation for each individual simulation of the age-depth model derived from the R-software Bacon v 2.4.3 (Blaauw and Christen, 2011). This results in a data-based frequency analysis that incorporates age-depth model uncertainties. The rather broad bandwidth is suited for showing general changes in frequency over time. To account for a higher resolution in frequency changes especially in periods with higher number of events, we also calculated a bandwidth based on the df occurrence of the last two centuries and applied the resulting 21-yr bandwidth to occurrence rate calculation of the main age. The cumulative thickness over time involves both the thickness and frequency of df turbidites and its slope provides information on df turbidite accumulation rate per year. We calculated the cumulative thickness on the mean values and the 95 % range values of the age-depth model to transfer age uncertainty to the cumulative thickness analysis."

>Lines 496-498
"The frequency analysis of the whole 4,000 year df turbidite record is based on a 150-yr bandwidth, whereas especially the higher number of events in the 20th century requires higher resolution frequency analysis, here on the basis of a 21-yr bandwidth (Fig. 8a)."

>Lines 526-530
"Df phase 4.1 is represented by a strong and fast frequency increase at ~1920 CE, followed by a period of highly frequent debris-flow events ~1980 CE. Since then, the current df phase 4.2 has lower frequencies relative to phase 4.1 but still by far higher frequencies than in the main df phases 1–3. Debris-flow frequency in 4.1 increased by a factor of 8 compared to the reference df phase 3. In df phase 4.2, debris-flow frequency increased by a factor of 7 compared to df phase 3."

(1) The used age-depth model by Oswald et. Al (2021) is primarily showing ages cal year BP.

(2) We changed the ages to CE/BCE in the revised supplement.

(1) Table S2 does not provide information on the uncertainties of the top layers or recent past.

(2) We added 95% range values of the age model for every debris-flow turbidite to the supplementary table S4 and stated the mean uncertainties in the text (Lines 510-511).

(3)

>Lines 510-511
"Mean age uncertainties are ± 6 years (2018-1960) and ±19 years (1959-1920; see also Supplementary Table S4)."

(1) On the one hand, I would expect the deviations from the mean age dating to be given for each identified event in table S4 and, on the other hand, I would find it methodologically very exciting how to deal with these uncertainties when the stratification of different drill cores of the events is known and overlapping by the dating-uncertainties? How to assign the identified df deposits to a specific year? At least in the recent past a validation with e.g. dendrochronological dating would have been useful?

(2) We appreciate the question on how to deal with absolute and relative age uncertainty. We address this problem by applying the frequency analysis on each of the 6,396 iterations of age-model simulation derived from Bacon instead of the mean event ages. The referee suggests to further calibrate the last decades by on-fan dendrochronological dating of the debris flows. We plan to include this technique in our future research to obtain terrestrial evidence and reduce age uncertainties for the lacustrine deposits. The inclusion of dendrochronological methods is, however, beyond the scope of the present study.

(1) This raises further the question whether all identified events can be used for a frequency analysis or for the classification of the four different phases based on different event rates?

(2) We address this comment in the revised discussion of human impact (Lines 636-656) and give more detailed information on human-induced vegetation changes based on pollen records. A period of enhanced forest clearance during medieval times (Kral 1989) is not reflected in a sudden increase in debris-flow frequency in the sediment core archive. Since then, no further increase of forest clearance or wildfires is reported in the area, so we infer that changes in vegetation can be neglected before or during the period of increasing debris-flow frequency in the last century (phase 4). We clarify in the new manuscript (Lines 649-650), that human induced mass wasting in the first half of the 20$^{th}$ century possibly created subaquatic detrital layers which cannot be distinguished from debris-flow deposits. This influence was considered in the frequency calculation (Lines 695-697).

Kral, F.: Pollenanalytische Untersuchungen im Fernpaßgebiet (Tirol): Zur Frage des Reliktcharakters der Bergsturz-Kiefernwälder, Verhandlungen der Zoologisch-Botanischen Gesellschaft in Wien. Since 2014 "Acta ZooBot Austria", 1989, 127–138, 1989.

(3)

>Lines 636-644
"Human-induced vegetation changes are documented since about 1,000 BCE from a pollen record from peat bog remnants at Heiterwanger See a few kilometres west of Plansee (Kral, 1989). A period of enhanced forest clearance in the area happened during medieval times according to this record. The herein presented sedimentary record of Plansee shows no signs of a drastic increase in debris-flow frequency throughout this period of intense forest use. During late medieval times, the climatic deterioration of the Little Ice Age, war, and epidemics led to a decrease in population and in forest clearance, which can be observed in pollen diagrams from the nearby Ostallgäu (Stojakowits and Friedmann, 2013). Since then, no further increase of forest clearance or wildfires is reported in the area. Therefore, we infer that there were no significant changes of vegetation before or during the period of increasing debris-flow frequency in the last century (df phase 4)."

>Lines 649-650
"Therefore, some of the df turbidites in the first half of the 20th century might actually reflect human-induced mass wasting (see also vi)."

(1) Is it possible to quantify the classification of the 4 phases by statistical test? However, once the uncertainties of the event frequency is known a trend analyses should be conducted, showing if changes in the occurrence rate or deposition rate/a are distinct or covered within the uncertainty estimates?

(2) We thank the referee for the important question on determining statistically robust changes. We opted for a change-point analysis of the inter-event periods using the R function *cpt.meanvar* with a clustering algorithm *BinSeg* following Albrecher et al. (2019) and references therein. The debris-flow

phases derived from the change-point analysis (Table 1) coincide with previously described lithotypes within the sediment cores (Lines 463-468). We added information on this in the methodology section (Lines 268-270) and in the results section (Lines 492-495).

Albrecher, H., Bladt, M., Kortschak, D., Prettenthaler, F., and Swierczynski, T.: Flood occurrence change-point analysis in the paleoflood record from Lake Mondsee (NE Alps), Global and Planetary Change, 178, 65–76, https://doi.org/10.1016/j.gloplacha.2019.04.009, 2019.

(3)

>Lines 268-270
"Phases with similar frequency of df turbidites (df phases) were statistically distinguished by a change-point analysis of the inter-event periods using the R function cpt.meanvar with a clustering algorithm BinSeg following Albrecher et al. (2019) and references therein."

>Lines 492-495
"Four main phases with similar frequency of df turbidites (df phases) are differentiated by a change-point analysis of inter-event periods of the 4,000 years debris-flow record (Fig. 8a, Tables 1, S2). Strikingly, the autonomously differentiated df phases temporally coincide with the above-described lithotypes indicating the major influence of debris flows to subaquatic event deposition in Plansee."

Specific minor comments:

(1) The shown correlation between precipitation and lake deposition is rather poor with respect to debris flow events, which can hardly be plausibly explained with precipitation data from a measuring station 7km away. Are there possibly better precipitation data (INAC?). However, the quality of the frequency series should be secured first.

(2) Potential mismatches between precipitation and debris-flow deposition might occur due to the age error of df turbidites, which we added in the new manuscript (Lines 683-685). Moreover, the data shown is daily precipitation, and debris flows can respond to hourly extremes. Of course precipitation occurs often locally concentrated in Alpine terrain. However, rainstorm activity is temporally concentrated coincident to certain meteorological conditions like Vb circulation systems, so there is a high correlation of nearby meteorological stations. Here we are even lucky enough to have a 100 year recording in 7 km distance, much better than in most similar studies that observe rainstorm frequencies over longer times scales in the Alps and local effects in rainstorm frequencies will also smooth out as an effect of the long observed period and the decadal view of rainstorm activities (see Lines 688-692).

(3)

>Lines 683-685
"Moreover, the precipitation sums are recorded on a daily base, where short but extreme convective precipitation are not distinguishable. Potential mismatches might also be due to the age error of df turbidites, which has a 95 % probability range from 5 years for the youngest event to 60 years at the begin of df phase 4."

(1) When considering earthquake activity in relation to the event rate presented, correlations become apparent and are also discussed. Although it is argued that only the strongest shaking events, well above the local intensity of VI, produce a distinct postseismic landscape response in a lacustrine record, Figure 6 (Supplementary) in Oswald et al. (2021) shows intensities above VI for the 1930

Namlos earthquake. Can an increase (if the time series is valid) after 1930 really be ruled out due to higher sediment availability caused by the earthquake?

(2) From our debris-flow record we deduce that debris-flow activity increases already at ~1920 CE. The possible minor catchment response for the 1930 CE earthquake with a local intensity of VI½ to VII at Plansee (Oswald et al., 2021) is in accordance to other lacustrine studies worldwide, where significantly enhanced fluvial sediment transport to lakes has only been observed for seismic intensities of ~VIII-IX (e.g Howarth et al., 2016; Moernaut et al., 2014). For the intensity VI-VII, the Environmental Seismic Intensity Scale (Michetti et al., 2007) also shows rather small landslide volumes. We therefore infer that the postseismic landscape response is not the dominant factor influencing debris-flow activity after 1930 at Plansee. We clarified this in our new manuscript (Lines 665-671).

Howarth, J. D., Fitzsimons, S. J., Norris, R. J., Langridge, R., and Vandergoes, M. J.: A 2000 yr rupture history for the Alpine fault derived from Lake Ellery, South Island, New Zealand, GSA Bulletin, 128, 627–643, https://doi.org/10.1130/B31300.1, 2016.

Moernaut, J., van Daele, M., Heirman, K., Fontijn, K., Strasser, M., Pino, M., Urrutia, R., and Batist, M. de: Lacustrine turbidites as a tool for quantitative earthquake reconstruction: New evidence for a variable rupture mode in south central Chile, Journal of Geophysical Research: Solid Earth, 119, 1607–1633, https://doi.org/10.1002/2013JB010738, 2014.

Michetti, A. M., Esposito, E., Roghozin, E., Guerrieri, L., and Porfido, S.: Environmental Seismic Intensity Scale - ESI 2007-, 2007.

(3)

>Lines 665-671
"The M 5.3 Namlos 1930 CE earthquake with local intensity of VI½ to VII at Plansee (Oswald et al., 2021) possibly had a minor influence on the sediment availability in the catchment. Given that debris-flow activity increases already at ~1920 CE, we infer that the postseismic landscape response after the 1930 CE earthquake is not the dominant factor influencing the debris-flow activity in the following decades. This is in accordance to other lacustrine studies worldwide, where significantly enhanced fluvial sediment transport to lakes has only been observed for seismic intensities of ~VIII-IX (e.g. Howarth et al., 2016; Moernaut et al., 2014)."